# Probabilistic Token Alignment for Large Language Model Fusion

**Runjia Zeng[1], James Chenhao Liang[2], Cheng Han[3], Zhiwen Cao[4], Jiahao Liu[5], Xiaojun Quan[6], Yingjie Victor Chen[7], Lifu Huang[8], Tong Geng[9,10], Qifan Wang[11], Dongfang Liu[1†]**

[1]Rochester Institute of Technology    [2]U.S. Naval Research Laboratory
[3]University of Missouri-Kansas City    [4]Adobe    [5]Meituan
[6]Sun Yat-sen University    [7]Purdue University    [8]UC Davis
[9]University of Rochester    [10]Rice University    [11]Meta AI    [†]Corresponding author

## Abstract

Training large language models (LLMs) from scratch can yield models with unique functionalities and strengths, but it is costly and often leads to redundant capabilities. A more cost-effective alternative is to fuse existing pre-trained LLMs with different architectures into a more powerful model. However, a key challenge in existing model fusion is their dependence on manually predefined vocabulary alignment, which may not generalize well across diverse contexts, leading to performance degradation in several evaluation. To solve this, we draw inspiration from distribution learning and propose the *probabilistic token alignment* method as a general and soft mapping for alignment, named as **PTA-LLM**. Our approach innovatively reformulates token alignment into a classic mathematical problem: *optimal transport*, seamlessly leveraging distribution-aware learning to facilitate more coherent model fusion. Apart from its inherent generality, PTA-LLM exhibits *interpretability from a distributional perspective*, offering insights into the essence of the token alignment. Empirical results demonstrate that probabilistic token alignment enhances the target model's performance across multiple capabilities. Our code is avaliable at runjia.tech/neurips_pta-llm.

## 1 Introduction

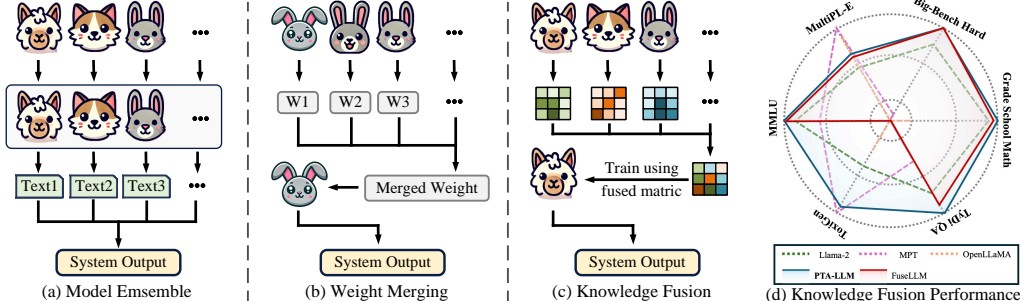

Figure 1: **PTA-LLM (ours)** *vs.* **concurrent arts** (*i.e.*, model ensemble [1] and weight merging [2]) under model fusion paradigm (*i.e.*, knowledge from the "cat" and "rabbit" source models is fused into the "llama" target model). Our knowledge fusion method yields general performance gains across multiple capabilities in (d), where all scores (see Table 2) are normalized for better visualization.

The rise of large language models (LLMs) such as Llama-2 [3], OpenLLaMA [4], and MPT [5], driven by scaling laws [6], has yielded significant advancement across a broad range of tasks (see Fig. 1 (d). The narrow dashed line area indicates its respective fields of advantage). Nevertheless, the reliance on scaling laws brings substantial computational demands [7], posing a noticeable

39th Conference on Neural Information Processing Systems (NeurIPS 2025).

impediment to the development of more robust baselines. A question thus naturally emerges: ① *How can we construct stronger baselines without resorting to the naive application of scaling laws?*

Pioneering research has begun to address this question through the concept of model fusion [8, 2, 9, 10], focusing on model ensemble (see Fig. 1 (a)) and weight merging (see Fig. 1 (b)). Recently, a prominent technique called knowledge fusion [11] aggregates the probabilistic distributions generated by individual LLMs and transfers this fused representation to a target model via distillation (see Fig. 1 (c)), enabling it to be more inference-efficient. After further employing token alignment [12], the misalignment issues arising from the use of different tokenizers across models are mitigated, allowing knowledge fusion to be architecture-agnostic. The exemplary advantages offered by knowledge fusion reframe question ① into ②: *How can we further optimize LLMs through knowledge fusion?*

Although knowledge fusion shows a promise avenue, two significant token alignment challenges still remain unresolved. ❶ The manually designed mapping strategy is overly simplistic, failing to capture the intricate patterns within the data. Tokens appearing in varying contexts often align with different objectives, and the bias introduced by this "rigid" alignment reduces the model's capacity to fully learn from the data, ultimately diminishing performance. ❷ The alignment of top-k predicted token sets from the source and target LLMs is performed independently, without taking into account their associated probabilities or overall distribution. This isolated strategy may achieve local optimality at each step alignment, but not a whole coherent fused metric. Thus, the core question ② becomes more specific: ③ *How can we effectively fuse LLMs with an adaptive and coherent matrix?*

To this end, we introduce **P**robabilistic **T**oken **A**lignment for **L**arge **L**anguage **M**odel Fusion (PTA-LLM). During the matrix fusion, we first employ dynamic algorithm to determine an optimal token pairing between the generated sequence from the source and target model. After obtaining the token pairings, a logit-level alignment will be conducted to resolve the token ID misalignment. Specifically, for the top-k predicted token sets from both source and target models, we hypothesize and further prove (see empirical results in Table 2 and 3) that the probabilistic distributions generated by distinct LLMs are coherent and reflective of their respective inherent knowledge. Therefore, PTA-LLM leverages the global generative distributions of each model's logits during token alignment, externalizing their collective knowledge and facilitating more precise mapping. To achieve this, our approach is grounded in Optimal Transport (OT), which optimally transforms one probability distribution into another while minimizing a predefined cost. By harnessing OT, we align or "transport" logit distributions between models, offering an effective solution. In contrast to hard mapping strategies, which align each token independently of its context, our proposed PTA-LLM employs a soft probabilistic alignment (detailed in §3.2). This approach better captures the intricacies of various linguistic context and thus establishes a stronger performance baseline, addressing the challenge ❶. Additionally, by incorporating distribution-aware learning, this method facilitates more consistent model representations (through the visualization results in §4.4), leading to marked improvements in generalization across a wide range of tasks (see Table §2), answering challenge ❷.

PTA-LLM enjoys a few attractive qualities. ***I. Generality.*** The global probabilistic distribution transport enhances the coherence of the representations, thereby improving the model's ability to generalize across a wide range of tasks and supporting the transfer of underlying representations for effective evaluation (see Table 2). ***II. Stability.*** The reframing through an optimal transport perspective introduces a soft probabilistic alignment, offering a flexible and adaptive solution to diverse contexts and performing stablly even in difficult tasks (see Table 3). ***III. Interpretability.*** The effectiveness of our approach is supported by theoretical insights from distribution learning and further validated through visualization results. It investigates the underlying mechanisms of token alignment, a critical operation in knowledge fusion that has been largely overlooked in prior research. This distinguishes PTA-LLM from most existing knowledge fusion models, which fail to elucidate precisely how token alignment works (see §4.4).

## 2 Related Work

**Model Fusion** has garnered significant attention as a means to enhance the general performance of LLMs. The fusion techniques can be categorized into three primary categories: *model ensembling*, *weight merging*, and *knowledge fusion*.

***Model ensembling*** combines the predictions of independently trained models to improve overall performance. Common approaches include weighted averaging [13], majority voting [14], and pairwise ranking [15]. However, a common challenge of model ensembling is that it requires

maintaining multiple models during inference, leading to high memory consumption and latency. ***Weight merging*** combines the parameters of multiple models to synthesize a new, unified model. This method is especially effective when the models share identical architectures [2, 9]. Weight merging is enhanced by linear mathematical operations on adapter parameters for improved model performance and generalization [16, 17, 18]. However, weight merging overly relies on architectural uniformity across models and requires manual tuning, limiting its applicability to diverse architectures (*i.e.*, low generalizability). In contrast, ***knowledge fusion*** offers a flexible and efficient model integration, particularly when the underlying architectures differ (*i.e.*, a common case in LLMs). It usually distills knowledge from multiple teacher models into a single student model. Based on its advantage, we follow this technique in our study. More discussions are presented in §S1.

**Token Alignment** was first introduced as a solution to address the misalignment problem between tokenizers with different size of vocabulary, specifically when aligning their respective distributions. The concept was initially formalized by [12], who employs a search algorithm to minimize the alignment cost between token sequences. Following research [11, 19] further explored flexible mappings for prominent performance via MinED strategy, statistical mapping, *etc*.

However, existing methods remain limited by their reliance on surface-level token correspondences (*i.e.*, based solely on the strings it comprises), which leverage minimum edit distance to align the logit. Our approach, besides using edit distance as one metric, advances token alignment by incorporating the corresponding logit values into the individual cost within the transport framework. Optimization is right now performed at both the "surface-level" and "logit-level" (see Eq. 4).

## 3 PTA-LLM

In this section, we present PTA-LLM, a novel probabilistic token alignment method for achieving general and coherent fusion of large language models (LLMs), as illustrated in Fig.2. Specifically, we outline our comprehensive knowledge fusion framework and tuning strategy in §3.1. Following this, we elaborate on the design of our probabilistic token alignment approach in §3.2, where the probabilistic distribution matrices from source LLMs are aligned into a fused representation via a dynamic pipeline, which involves two primary stages: *dynamic token pairing* and *probabilistic token alignment*. Last but not least, in §S2, we provide a detailed description of the implementation and the algorithm utilized in our approach. More implementation details will be provided in §S5.

Table 1: The key notations used in PTA-LLM. See all notations in §S4

| Notation | Representation |
|---|---|
| $\mathcal{C}$ | A training corpus consisting of a collection of text sequences $t$ |
| $\mathcal{P}_s$ | The probabilistic distribution matrix for the source model, consisting of $L$ tokens (denoted as $\mathcal{A} \in \mathbb{R}^{V_s}$) |
| $\mathcal{P}_t$ | The probabilistic distribution matrix for the target model, consisting of $N$ tokens (denoted as $\mathcal{B} \in \mathbb{R}^{V_t}$) |
| $\mathcal{P}_f$ | The fused probabilistic distribution matrices for model fusion, consisting of $N$ tokens (denoted as $\hat{\mathcal{B}} \in \mathbb{R}^{V_t}$) |
| $\mathbf{Q}_t$ | Target model's predictions during the fine tuning |
| $\hat{\mathcal{T}}$ | The final $n \times m$ optimal transport plan matrix of non-negative entries $\mathcal{G}_{nm}$ |

### 3.1 Problem Statement & Overall Objective

Let $t$ represent an input text sequence sampled from a corpus $\mathcal{C}$. A probabilistic distribution matrix $\mathcal{P} \in \mathbb{R}^{K \times V}$ is obtained by evaluating the output token distirbutions of a large language model (LLM) from the input $t$, where $K$ corresponds to the output sequence length, and $V$ denotes the size of the vocabulary. The $i$-th row of this matrix represents the predicted probability distribution over the vocabulary for the $i$-th token in the sequence. In the context of combining two LLMs (source and target), we consider the probabilistic distribution matrices $\mathcal{P}_s \in \mathbb{R}^{L \times V_s}$ for the source model and $\mathcal{P}_t \in \mathbb{R}^{N \times V_t}$ for the target model, where $L$ and $N$ denote the sequence lengths, and $V_s$ and $V_t$ represent the vocabulary sizes of the source and target models, respectively. *When these models employ different tokenization schemes, misalignment between the tokens of the source and target models arises, thereby complicating the integration of their probabilistic outputs.* Addressing this issue is essential for effectively combining the outputs of both models. The traditional approach seeks to ensure consistency between the target model's predictions, denoted as $\mathbf{Q}_t$, and the fused representation $\mathcal{P}_f$, which encapsulates the knowledge from the source model. The knowledge fusion loss is formulated as $\mathcal{L}_{\text{Fusion}} = -\mathbb{E}_{t \sim \mathcal{C}} \left[ \mathbb{D}(\mathbf{Q}_t, \mathcal{P}_f) \right]$, where $\mathbb{D}(\cdot, \cdot)$ is a discrepancy function (such as cross-entropy or KL divergence) measuring the difference between the predicted and fused probability

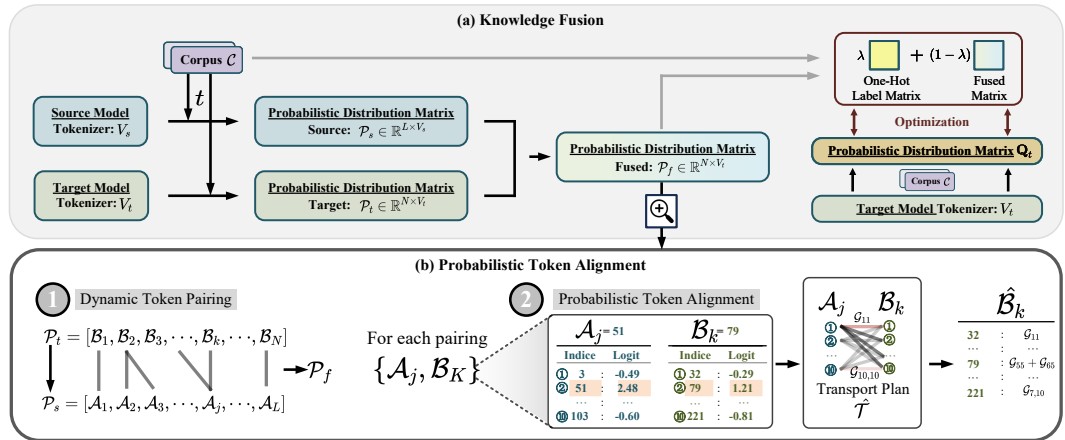

Figure 2: **Probabilistic token alignment under the knowledge fusion paradigm.** (a) The overall knowledge fusion pipeline (see §3.1), and (b) two-stage probabilistic token alignment (see §3.2), including dynamic token pairing and probabilistic alignment using optimal transport reformulation.

distributions. The fused output $\mathcal{P}_f$ is a probabilistic distribution matrix that represents the combined strengths of both the source and target models, formally defined as $\mathcal{P}_f = \text{MatrixAlignment}(\mathcal{P}_s, \mathcal{P}_t)$.

In this work, we propose PTA-LLM, a framework designed to resolve discrepancies between the tokenization schemes of the source and target models. The principal objective is to minimize the divergence between the target model's probabilistic predictions $\mathbf{Q}_t$ and the corresponding one-hot encoded label matrix $\mathbf{O}_t \in {0,1}^{N \times V}$, where each row of $\mathbf{O}_t$ indicates the correct token as a one-hot vector. Specifically, we define a causal language modeling (CLM) loss, which measures this divergence, as $\mathcal{L}_{\text{CLM}} = -\mathbb{E}_{t \sim \mathcal{C}}\left[\mathbb{D}(\mathbf{Q}_t, \mathbf{O}_t)\right]$, between the predicted probabilities and the true labels. Consequently, the overall training objective of our proposed method is to optimize a weighted combination of the CLM loss and the fusion loss, formalized as $\mathcal{L} = \lambda\mathcal{L}_{\text{CLM}} + (1 - \lambda)\mathcal{L}_{\text{Fusion}}$, where $\lambda \in [0, 1]$ is a hyperparameter controlling the trade-off between the causal language modeling loss and the fusion objective. This ensures that the target model can effectively learn from both its own predictions and the knowledge transferred from the source model.

## 3.2 Probabilistic Token Alignment

**Dynamic Token Pairing**   The task of aligning two distinct probabilistic distribution matrices, $\mathcal{P}_s$ and $\mathcal{P}_t$ poses a significant computational challenge due to the inherent differences in both sequence length and vocabulary size. The core problem involves finding a suitable alignment between tokens from the source model's distribution $\mathcal{P}_s$ and those from the target model's distribution $\mathcal{P}_t$. More precisely, for each token $\mathcal{A}_j$ ($j \in [1, L]$) from $\mathcal{P}_s$, we aim to pair it with a corresponding token $\mathcal{B}_k$ ($k \in [1, N]$) from $\mathcal{P}_t$ as shown in Fig.2 (b1).

Given that there are $L \times N$ potential pairings between these tokens, employing brute-force attempts to explore all possible combinations would be computationally prohibitive, especially as the sequence lengths and vocabulary sizes grow. To address this, we introduce dynamic token pairing, which provides an efficient way to systematically explore the space of possible pairings and compute an optimal alignment. This approach allows for the minimization of computational complexity while ensuring the best mapping between the source and target tokens.

Formally, given two sequences of tokens $[\mathcal{A}_{1:L}, \mathcal{B}_{1:N}]$, our objective is to find an alignment that minimizes the overall cost associated with transforming one sequence into the other. Thus, we define the recursion function as:

$$f(k, j) = \min\{f(k - 1, j) + c(\mathcal{B}_k, \mathcal{A}_j),$$
$$f(k, j - 1) + c(\mathcal{B}_k, \mathcal{A}_j), \tag{1}$$
$$f(k - 1, j - 1) + c(\mathcal{B}_k, \mathcal{A}_j)\}, k \in [1, N], j \in [1, L]$$

where $f(k, j)$ represents the total cost of aligning the subsequences $\mathcal{B}_{1:k}$ and $\mathcal{A}_{1:j}$, while $c(\mathcal{B}_k, \mathcal{A}_j)$ denotes the predefined cost or distance metric between tokens. In contrast to traditional alignment

methods [20, 21], which typically enforce a one-to-one correspondence between elements in the two sequences, our approach introduces generality by relaxing this constraint. Specifically, our formulation allows the dynamic possibility assignment that one token in the source domain may align with multiple tokens in the target domain, and vice versa, depending on the characteristics of the tokenization schemes and the specific demands of the alignment task.

By adopting this dynamic token paring strategy, our method is able to handle discrepancies between the tokenization schemes of the source and target models, ensuring that the probabilistic distributions $\mathcal{P}_s$ and $\mathcal{P}_t$ can be meaningfully aligned, even in cases where their underlying token structures differ significantly. This enhanced flexibility is particularly useful in scenarios where the vocabulary sizes and token sequences vary substantially. Ultimately, we provide a more robust solution to the alignment problem in the context of knowledge fusion between models.

**Probabilistic Token Alignment**     After determining the optimal token pairings, the next fundamental step involves accurately performing logit-level alignment to address the token ID (*i.e.*, we transform the text into corresponding token IDs using a tokenizer.) misalignment that arises due to the use of different tokenization schemes. As shown in Fig. 2 (b2) , even when $\mathcal{A}_j$ and $\mathcal{B}_k$ are decoded into the same text by their respective tokenizers, their token ID may differ (*i.e.*, 51 *vs.* 79). Specifically, for each token pair $\mathcal{A}_j \in \mathbb{R}^{V_s}$ and $\mathcal{B}_k \in \mathbb{R}^{V_t}$, the objective of token alignment is to match the logits from the source token with the logits from the target token in order to achieve consistent representations between the models. The resulting fused token distribution, denoted as $\hat{\mathcal{B}}_k$, can be defined as:

$$\hat{\mathcal{B}}_k = \text{TokenAlignment} \left( \mathcal{A}_j, \mathcal{B}_k \right), \tag{2}$$

where $\text{TokenAlignment}$ is a function that fuses the logits from the source and target models for each token pairing. This fusion process aims to produce a unified token distribution by combining the outputs from both the source and target language models. In addition, Equation 2 highlights that the token fusion for each pairing can be reformulated from the perspective of distribution learning, where the goal is to minimize discrepancies between the two token distributions. Formally, we have:

$$\hat{\mathcal{T}} = \arg\min \mathcal{L} \left( \mathcal{A}_j, \mathcal{B}_k \right), \tag{3}$$

where the loss function $\mathcal{L}$ represents the information loss incurred during the alignment process. The goal is to minimize this loss, ensuring that the information from the source logits is effectively transferred to the target logits without significant degradation.

This optimization problem is conceptually analogous to the classical problem of optimal transport. Our objective is to find a "transport plan" $\hat{\mathcal{T}}$ that minimizes the total cost of transferring probability mass from one distribution, $\mu$, to another distribution, $\nu$. Hence, in the context of token alignment, we can reinterpret the task as an Optimal Transport (OT) problem, where the aim is to determine a global transport plan that transfers the logits of the source tokens $\mathcal{A}_j$ to the logits of the target tokens $\mathcal{B}_k$ at minimal cost. This process, under our setting, is formulated as:

$$\hat{\mathcal{T}} = \arg\min_{\mathcal{T} \geq 0} \left\{ \sum_{x=1}^{n} \sum_{y=1}^{m} c_{xy} \, \mathcal{G}_{xy} \, \middle| \, \sum_{y=1}^{m} \mathcal{G}_{xy} = \mathcal{A}_j[x] \forall x, \quad \sum_{x=1}^{n} \mathcal{G}_{xy} = \mathcal{B}_k[y] \forall y \right\}, n = m = 10, \tag{4}$$

where $\hat{\mathcal{T}}$ is an $n \times m$ matrix of non-negative entries $\mathcal{G}_{xy}$, representing the amount of logit probability transported from the $x$-th indice in the source token $\mathcal{A}_j$ to the $y$-th indice in the target token $\mathcal{B}_k$. The cost matrix $C$ captures the alignment cost between source token $\mathcal{A}_j$ and target token $\mathcal{B}_k$, where we define $c_{xy}$ as the minimum edit distance (after decoding the indice into text) between the $x$-th indice in the source token $\mathcal{A}_j$ and the $y$-th indice in the target token $\mathcal{B}_k$ (*i.e.*, the $\mathcal{L}$ in Equation 3). The constraints $\sum_{y=1}^{m} \mathcal{G}_{xy} = \mathcal{A}_j[x]$ and $\sum_{x=1}^{n} \mathcal{G}_{xy} = \mathcal{B}_k[y]$ ensure "logit probability" conservation between the source and target token distributions. See more details in §S5.

Once the "transport plan" $\hat{\mathcal{T}}$ is determined, the next step is to align the logits by selecting the target token logits with the highest probability for each source token logit, which can be reformulated as:

$$\hat{\mathcal{B}}_k = \left\{ (r, \mathcal{G}_{xy}) \, \middle| \, r \in R_x \right\}. \tag{5}$$

Here each pair consists of the index $r$ and the corresponding transport probability $\mathcal{G}_{xy}$ from the optimal transport plan $\hat{\mathcal{T}}$. The set $R_x$ represents the indices corresponding to the largest values in

the $x$-th row of $\hat{\mathcal{T}}$, which indicate the most probable target token logits for alignment with the $x$-th source token logit. As shown in Fig. 2 (b2), the final fused logits for each indice in $\hat{\mathcal{B}}_k$ are determined by the maximum transported logits. Notably, if multiple source token indices contribute the highest transported logits to the same target indice, their contributions are accumulated (*i.e.*, $\mathcal{G}_{55} + \mathcal{G}_{65}$). We demonstrate that our probabilistic token alignment can generate an more adaptive (see empirical results in Table 2) and coherent (see the visualization of token in §4.4) fused matrix.

### 3.3 Implementation Detail

In this section, we present the implementation details of optimal transport and the fusion strategy for fusing different LLMs in our PTA-LLM method.

**Optimal Transport**    As stated in Equation 4 and 5, the token alignment tasks are transformed into OT problem. Consequently, how to efficiently compute the global transport plan becomes crucial. To address this, we employ the Sinkhorn algorithm [22] to solve the optimal transport problem following common practice [23]. The implementation of Sinkhorn algorithm is shown in Algorithm 1.

---

**Algorithm 1** Sinkhorn Algorithm for Optimal Transport

---

**Require:** Cost matrix $C$, source token distribution $\mathcal{A}_j$, target token distribution $\mathcal{B}_k$, temperature $\lambda$
1: Initialize $\mathcal{T} = \exp\left(-\lambda C\right)$
2: **repeat**
3:     scale the rows of $\mathcal{T}$ such that the row sums match $\mathcal{A}_j$
4:     scale the columns of $\mathcal{T}$ such that the column sums match $\mathcal{B}_k$
5: **until** convergence
6: **return** $\hat{\mathcal{T}}$.

---

**Fusion Strategy**    To effectively merge the collective knowledge of source LLMs while retaining their individual strengths, it's crucial to assess the quality of each LLM and assign different levels of importance to their respective distribution matrices. To do this, when processing text $t$, we employ cross-entropy loss between the distribution matrices and the gold labels as a measure of the LLMs' prediction quality [24]. A lower cross-entropy score for a source LLM indicates a more accurate understanding of the text, and its prediction should thus be given greater weight. Following this principle, we select the distribution matrix with the lowest cross-entropy score as the source LLM distribution matrix. More fusion strategy ablative studies results are shown in Table 4b

## 4 Experiments

### 4.1 Experimental Setup

**Training details**  We fine-tune the Llama-2 7B model using a batch size of 256 and a maximum sequence length of 2,048 tokens with a combination weight (*i.e.*, the $\lambda$ in §3.1 ) of 0.8 on MiniPile [25] following [11]. More details are presented in §S2.
**Evaluation**  We evaluate PTA-LLM on six benchmarks (see details in §S3) that span various core capabilities of LLMs, including *reasoning*, *coding*, *commonsense*, *safty* and *multilingual ability*.
**Baselines**  We evaluate the performance of PTA-LLM with three sets of baselines: (1) *Source LLMs*, including Llama-2 7B [3], OpenLLaMA 7B [4], and MPT 7B[5]; (2) *Llama-2 CLM*, a Llama-2 7B model that further fine tuned on MiniPile using the traditional causal language modeling objective; and (3) FUSELLM [11], a Llama-2 7B model trained on MiniPile with an emphasis on integrating the capabilities of multiple source models under the knowledge fusion paradigm.
**Reproducibility**  PTA-LLM is implemented in Pytorch [26] using the Huggingface Transformers library [27], accelerated by FlashAttention [28]. Our full implementation will be publicly released.

### 4.2 Main Results

Table 2 presents the overall performance of PTA-LLM compared to three sets of baseline models (*i.e.*, source LLMs, Llama-2 CLM and FUSELLM). The results indicate that the original LLMs exhibit varying performance across the six benchmarks, with Llama-2 generally achieving the best results, while MPT demonstrates the weakest overall performance. Following continual training on MiniPile, Llama-2 CLM shows a modest average improvement of 1.20% over the original Llama-2 model.

Table 2: Overall results of PTA-LLM and baselines in six various benchmarks, including 78 tasks in total. The percentages indicate the rate of improvement/decrease compared to FUSELLM. We further report "Number of Tasks" in [·]. Notably, higher average values indicate better performance in each benchmark. Per-task results and more experiment details are available in Appendix §S8.

| Benchmark [# of Tasks] | | OpenLLaMA | MPT | Llama-2 | Llama-2 CLM | FUSELLM | PTA-LLM |
|---|---|---|---|---|---|---|---|
| Grade School Math | [1] | 7.81 | 9.17 | 14.18 | 14.33 | 14.56 | **14.71** (+1.03%) |
| Big-Bench Hard | [27] | 33.87 | 33.38 | 39.70 | 40.44 | 41.01 | **41.08** (+0.17%) |
| MultiPL-E | [10] | 18.11 | 17.26 | 14.63 | 14.83 | 15.56 | **15.88** (+2.06%) |
| MMLU | [17] | 42.11 | 27.84 | 46.94 | 47.65 | 48.77 | **49.38** (+1.25%) |
| ToxiGen | [14] | 18.94 | 18.42 | 18.56 | 18.33 | 18.19 | **18.89** (+3.85%) |
| TyDi QA | [9] | 27.32 | 22.11 | 31.42 | 31.80 | 32.99 | **34.07** (+3.27%) |
| Avg. 6 Benchmarks | [78] | 24.69 | 21.36 | 27.57 | 27.90 | 28.51 | **29.00** (+1.72%) |

Compared to FUSELLM, PTA-LLM demonstrates an average relative performance gain of 1.72% across 78 tasks. Notably, in the challenging benchmark of ME, which consists of multiple popular programming languages, our approach achieves a significant performance gain of +2.06% compared with Llama-2. Notable improvements are also observed in core areas such as *safety* and *multiling*. While a slight performance degradation is observed in the continual training for the ToxiGen benchmark under FUSELLM, PTA-LLM achieves a 3.85% relative improvement, highlighting the generality of probabilistic token alignment across diverse contexts. We also find that PTA-LLM experiences a minor performance improvement (*i.e.*, +0.17%) on the BBH benchmark compared to FUSELLM. This decline can be attributed to poor performance of source models. Two of the three source models (*i.e.*, OpenLLaMA and MPT) underperform on these tasks, and thus their more coherent token alignment may inadvertently hinder continual training effectiveness in a reasonable jitter. In conclusion, PTA-LLM improves the model's ability to generalize across a wide range of tasks and supports the transfer of underlying representations for effective evaluation.

## 4.3 Study of Stability

Table 3: Case study of PTA-LLM in the performance degradation tasks for continue training and FUSELLM. The percentages indicate the rate of improvement/decrease compared to Llama-2. We also denotes its corresponding benchmark in [·]. Case studies for BBH are provided in §S9.

| Task [Benchmark] | Llama-2 | Llama-2 CLM | FUSELLM | PTA-LLM |
|---|---|---|---|---|
| Causal Judgement [BBH] | 50.80 | 46.52 (-8.43%) | 46.52 (-8.43%) | **50.80** (+0.00%) |
| Geometric Shapes [BBH] | 34.40 | 19.20 (-44.17%) | 22.80 (-33.72%) | **26.80** (-22.09%) |
| Tracking Shuffled Objects (7 objects) [BBH] | 11.20 | 9.60 (-14.29%) | 10.40 (-7.14%) | **14.00** (+25.00%) |
| Chemistry [MMLU] | 35.97 | 34.11 (-5.17%) | 34.98 (-2.75%) | **36.96** (+2.75%) |
| Jewish [ToxiGen] | 27.00 | 21.60 (-20.00%) | 23.80 (-11.85%) | **25.20** (-6.67%) |
| Arabic [TyDi QA] | 8.47 | 5.45 (-35.66%) | 5.65 (-33.29%) | **7.49** (-11.57%) |
| Swahili [TyDi QA] | 43.69 | 38.97 (-10.80%) | 39.78 (-8.95%) | **41.68** (-4.60%) |
| Avg. 7 Tasks | 30.22 | 25.06 (-17.07%) | 26.28 (-13.04%) | **28.99** (-4.07%) |

We observe that in certain tasks (6 out of 43 tasks), FUSELLM under the knowledge fusion paradigm exhibits performance degradation, which significantly diminishes its overall efficacy. This suggests instability when exposed to perturbations, such as more challenging or unseen tasks. Consequently, a thorough analysis of these tasks is necessary to provide valuable insights for future research.

Our hypothesis is that the hard mapping token alignment strategy employed by FUSELLM is suboptimal in these contexts, necessitating manual specification of alignment strategies tailored to each task for improved outcomes. In contrast, our method reframes the problem through the perspective of optimal transport, introducing a soft probabilistic alignment that offers greater flexibility and adaptability across diverse tasks. This approach not only mitigates performance degradation (*i.e.*, achieve an overall 8.97% performance mitigation over FUSELLM) but also results in significant improvements, particularly in benchmarks such as BBH (*i.e.*, **14.00** *vs.* 11.20) and MMLU (*i.e.*, **36.96** *vs.* 35.97). For instance, our method achieves a 25.00% improvement over Llama-2 in the tracking shuffled objects task. These promising results underscore the stability of probabilistic token alignment in enhancing model performance across varied contexts.

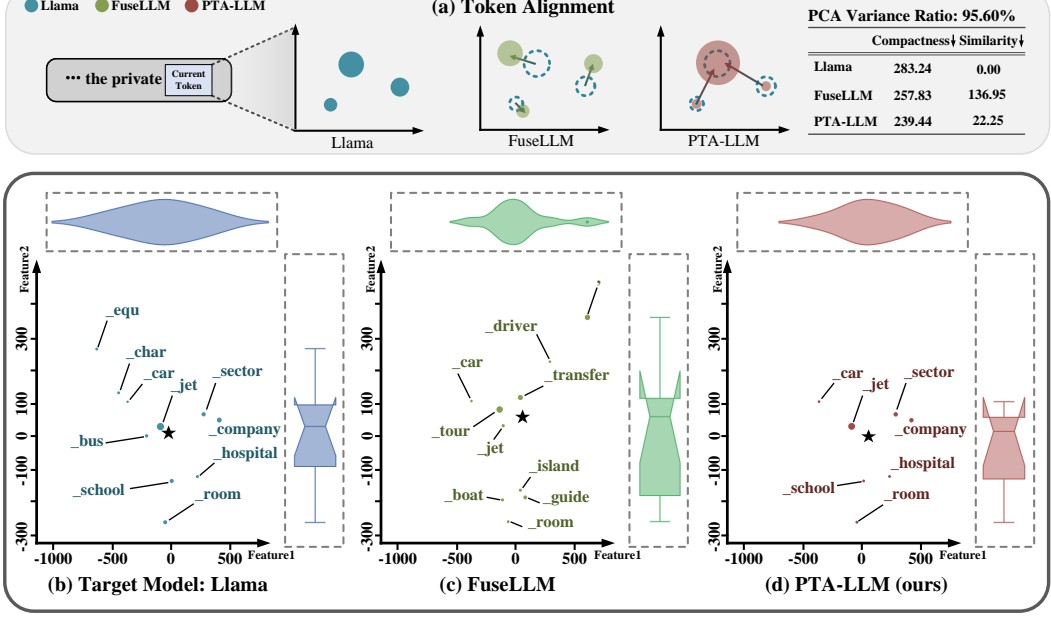

Figure 3: **Study of Interpretability.** (a) The abstract understanding of token alignment in FUSELLM and PTA-LLM and their respective evaluation metrics. (b) 2D visualization results of target tokens and fused tokens, where their locations represent semantic information and the sizes indicate their corresponding logit magnitudes. The ⋆ on the coordinates denotes the logit-weighted center of each token. Additional visualization results are presented in §S6.

### 4.4 Study of Interpretability

Although the emergence of knowledge fusion as a model fusion paradigm has gained huge attention, the underlying rationale remains unclear. In this section, we tend to provide distribution insights into token alignment's mechanisms and offer guidance for its optimal utilization. As shown in Fig. 3, we delve into a specific context to have an in-depth analysis of token alignment. Given we have previously aligned tokens like "the private," we need to align the token pair from the source model and target model to form the next fused token. For tokens from the target model (*i.e.*, Llama-2), we can visualize their top-10 logits and corresponding indices in a 2D space (see Fig. 3 (a), left coordinate, showing only 3 logits in a high-level representation). This is done by first using the target model's tokenizer to extract token features, followed by dimensionality reduction using Isomap [29] and PCA [30] (the variance ratio is reported as 95.60% on average in the table in Fig. 3 (a)). Their relative position can reflect the underlying meaning of this indice, and the relative size indicates the magnitude of their corresponding logit. For FUSELLM, traditional hard mapping does not consider their logit and maps each indice to another with a pre-defined strategy, acting like a "moving" (*i.e.*, change the location without modifying the size) in high-level understanding. In contrast, our method leverages the complete distribution, "transporting" (*i.e.*, distribute the size into current location) the optimal logit into existing indices. Quantitatively, we further compute the average compactness of each token (*i.e.*, the logit-weighted Euclidean distance from each point to its center) and the similarity of each token center to the target one (*i.e.*, the Euclidean distance from each center point to the target one) in 100 random samples, as shown in the table in Fig. 3 (a). It empirically demonstrates that our method generates a more coherent fused token, as evidenced by a more compact representation (*i.e.*, lower inner distance: 239.44 *vs.* 257.83) and a more consistent representation (*i.e.*, lower center distance: 22.25 *vs.* 136.95).

As shown in the down part of Fig. 3, we can visually compare the distribution of PTA-LLM fused token with the target token and FUSELLM fused token. Specifically, a more consistent marginal feature distribution between PTA-LLM and target token can be observed from Fig. 3 (b) and Fig. 3 (d), where FUSELLM exhibits significantly greater distortion in the overall token representation. The more compact and coherent overall token distribution after employing probabilistic token alignment is aligned with the quantitative results. More implementation details will be elaborated in §S6.

Table 4: A set of **ablative studies** on three different core capablities evaluation benchmarks (*i.e.*, BBH, MMLU, ME). (a) The probabilistic token alignment parameters include two key hyperparameters: convergence threshold and transport window size. (b) The fusion training parameters consist of the combination weight, which controls the relative emphasis during continued training, while the fusion function determines the source distribution matrix at each training step. See more results in §S10

| Choice | BBH | ME | MMLU | Choice | BBH | ME | MMLU |
|--------|-----|-----|------|--------|-----|-----|------|
| *Optimal Transport Convergence Threshold* | | | | *Combination Weight* | | | |
| 1e-4 | 40.54 | 15.88 | 48.99 | 0.9 | 40.39 | 15.72 | 48.93 |
| 1e-5 | 41.08 (+1.33%) | 15.82 (-0.38%) | 49.38 (+0.80%) | 0.8 | 41.08 (+1.71%) | 15.88 (+1.04%) | 49.38 (+0.92%) |
| *Token Alignment Window Size* | | | | *Fusion Function* | | | |
| 10 | 41.08 | 15.88 | 48.99 | AvgCE | 40.52 | 15.69 | 48.89 |
| 5 | 40.68 (-0.97%) | 15.61 (-1.70%) | 49.38 (+0.78%) | MinCE | 41.08 (+1.38%) | 15.88 (+1.23%) | 49.38 (+1.00%) |

| (a) Probabilistic Token Alignment Parameters. | (b) Fusion Trainning Parameters |
|---|---|

## 4.5 Diagnostic Experiment

**Number of source LLMs.** In Table 5, we present the results of fusing varying numbers of LLMs. In general, the performance of PTA-LLM improves as the number of integrated models increases from 1 to 3. However, we also find that the benefits of incorporating additional models vary across different benchmarks (*i.e.*, a prominent improvement is observed in the ME). It is also important to highlight that the fusion of

Table 5: Results of PTA-LLM by incorporating varying numbers (from 1 to 2) of models.

| Model | BBH | MMLU | ME |
|-------|-----|------|-----|
| OpenLLaMA | 33.87 | 42.11 | 18.11 |
| MPT | 33.38 | 27.84 | 17.26 |
| Llama-2 | 39.70 | 46.94 | 14.63 |
| Llama-2 CLM | 40.44 (+1.86%) | 47.65 (+1.51%) | 14.83 (+1.37%) |
| Llama-2 + OpenLLaMA | 40.54 (+2.11%) | 49.26 (+4.95%) | 15.83 (+8.17%) |
| Llama-2 + MPT | 40.65 (+2.39%) | 48.19 (+2.67%) | 15.78 (+7.88%) |
| PTA-LLM | 41.08 (+3.48%) | 49.38(+5.20%) | 15.88 (+8.54%) |

lower-performing source models results in diminished performance gains (*i.e.*, MPT, which performs the worst in the MMLU benchmark, contributes the least improvement when we combine one model).

**Optimal Transport Convergence Threshold.** As discussed in §S2, a key hyperparameter in optimal transport is the threshold, which regulates the convergence of the Sinkhorn algorithm [22]. A lower value of threshold results in more iterations of transport, enforcing a stricter distribution constraint. As illustrated in Table 4a (up), the lower optimal temperature preference indicates that a stricter constraint may form a more coherent fusion and thus bring a greater performance gain.

**Token Alignment Window Size.** During the probabilistic token alignment, the default transport window size is the same of the logit length (*i.e.*, Top-10). Here, we explore the impact of window size on the transport of fused logit in Table 4a (down). In general, larger transport range enable a more comprehensive understanding of the context and thus lead to a performance improvement.

**Combination Weight.** As discussed in §3.1, the combination weight determines the relative emphasis placed on the fused matrix versus the label matrix during continued training. We can observe a higher performance in Table 4b (up) when the weight is smaller within a reasonable range (see detailed analysis in §S10), since a lower value indicates more emphasis in our fused matrix.

**Fusion Functions.** In §S2, we employ a distribution matrix with minimum cross entropy (MinCE) to define the source distribution matrix during training. Additionally, we implement a weighted average of distribution matrices based on cross entropy (AvgCE). A comparison of these two approaches is provided in Table 4b (down). The results show that PTA-LLM using MinCE consistently outperforms AvgCE across all benchmarks, which is consistent with [11].

## 5 Conclusion

We present **P**robabilistic **T**oken **A**lignment for **L**arge **L**anguage **M**odel Fusion (**PTA-LLM**), a distribution-wise token alignment approach that leverages the optimal transport framework through reformulation. It has merits in: **i)** demonstrating generality across benchmarks through a coherent representation fusion; **ii)** offering a flexible and adaptive solution to various contexts, especially stable in addressing challenging tasks; and **iii)** thoroughly investigating the essence of token alignment to elucidate the coherent token we fused. As a whole, we conclude that the outcomes elucidated in this paper impart essential understandings and necessitate further exploration within this realm.

## 6 Acknowledgements

This research was supported by the National Science Foundation under Grant No. 2242243, 2348468 & 2450068.

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

# S1    Related Work

**Model Fusion** has garnered significant attention as a means to enhance the general performance of LLMs. The fusion techniques can be classified into three primary categories: *model ensembling*, *weight merging*, and *knowledge fusion*. *Model ensembling* combines the predictions of independently trained models to improve overall performance. Common approaches include weighted averaging [31], majority voting [1], and pairwise ranking [15]. Although model ensembling often leads to significant improvements in predictive accuracy and model robustness, it requires maintaining multiple models during inference, leading to higher memory consumption and increased latency. This makes it less efficient for resource-constrained environments. *Weight merging* combines the parameters of multiple models to synthesize a new, unified model. This method is especially effective when the models share identical architectures, as their parameters can be merged seamlessly [2, 9]. Weight merging is enhanced by linear mathematical operations on adapter parameters, which has proven useful for improving model performance and generalization [16, 17, 18]. Despite these advantages, weight merging suffers from significant limitations: It relies on architectural uniformity across models and requires manual tuning, which constrains its applicability across diverse model architectures (*i.e.*, low generalizability).

In contrast, *knowledge fusion* offers a more flexible and efficient means of integrating models, particularly when the underlying architectures differ (*i.e.*, a common case in LLMs). It distills knowledge from multiple teacher models into a single student model, transferring the knowledge in a more compact and efficient form. One of the key innovations is the minimum edit distance (MinED) token alignment strategy, first introduced by [11], which facilitates effective knowledge transfer by aligning tokens across models. This approach was further refined by [19], who proposed a mapping statistics-based strategy designed to enhance conversational model performance. Compared to model ensembling and weight merging, knowledge fusion presents a more scalable and architecture-agnostic solution, making it highly suitable for integrating multiple LLMs while minimizing the performance degradation typically associated with stepwise optimization.

**Token Alignment** was first introduced as a solution to address the misalignment problem between tokenizers with different size of vocabulary, specifically when aligning their respective distributions. The concept was initially formalized by [12], who employs a search algorithm to minimize the alignment cost between token sequences. This method relies on the assumption that an optimal one-to-one mapping between tokens can be found, enabling the direct alignment of their respective distributions. However, in cases where such a precise mapping is not feasible, the solution defaults to a one-hot vector representation, which may oversimplify the complexities inherent in real-world token distributions. Building upon this work, [11] introduced a more flexible approach by replacing the exact match requirement with MinED strategy for more robust token alignment, especially in cases where slight variations between tokens could still preserve semantic equivalence. Later, [19]

refined further in cross-lingual applications, incorporating statistical mapping frequencies between source and target tokens to better account for the probabilistic nature of token co-occurrence, leadning to a prominent chat performance.

However, existing methods remain limited by their reliance on surface-level token correspondences (*i.e.*, based solely on the strings it comprises), which leverage minimum edit distance to align the logit. However, besides using edit distance as one metric, our method advances this by incorporating the corresponding logit values into the individual cost within the transport framework. Optimization is performed at both the "surface-level" and "logit-level".

## S2    Implementation Details

In this section, we present the relevant training details for fusing different LLMs in our PTA-LLM.

**Training Time**    Training is conducted on 8 NVIDIA A100-80GB GPUs (approximately 26 hours for a single epoch) and 8 NVIDIA H100-80GB GPUs (approximately 17 hours for a single epoch), while conducting evaluation on 4 NVIDIA A100-40GB GPUs (time varies depending on the amount of benchmark data used).

More specifically, the entire runtime analysis can be divided into Stage 1 and Stage 2.

Stage 1 (Training-free). For LLaMA-CLM, no token alignment is required, so the runtime for this stage is zero. For FuseLLM, Stage 1 takes approximately 3 GPU hours (on NVIDIA A100-80GB) and 4 CPU hours (on AMD EPYC 7763 processor). Compared to FuseLLM, PTA-LLM introduces an additional optimal transport computation, which results in approximately a 13.75% increase in CPU alignment time on the MiniPile dataset. It would be valuable to explore more efficient alternatives, such as the Greenkhorn algorithm, to reduce the complexity of the optimal transport step.

Stage 2 (Training stage). We report the runtime for PTA-LLM (*i.e.*, end-to-end training) under various settings (*e.g.*, different GPUs and dataset sizes) in §S7. Since Stage 2 only involves standard end-to-end training, the corresponding runtime for FuseLLM and LLaMA-CLM are comparable under the same condition

**Training Results**    While our average absolute gain is 1.1% compared to the CLM, the relative gain of 3.94% is nontrivial. In addition, our significance test results indicate that the improvements are statistically significant with p-value < 0.005. To assess the stability of our results, we conducted three independent runs using identical hyperparameter settings for each benchmark. The standard deviations of the evaluation scores were 0.05 for MMLU, 0.04 for BBH, and 0.05 for MultiPL-E. Our findings indicate that the performance remains stable once the hyperparameters and training devices are fixed. Other benchmarks show similar trends on standard deviation. We have included these results in the revised paper.

**Training Dataset**    MiniPile is a compact yet diverse training dataset consisting of up to 1 million samples, carefully curated from the Pile to preserve the original corpus's richness across various domains while maintaining a manageable size for efficient experimentation.

**Base Model Tokenizers**    Both LLaMA-2, developed by Meta AI, and OpenLLaMA, an open-source reproduction of LLaMA, utilize tokenizers with a vocabulary size of 32,000 tokens. These two models share a substantial overlap, with 20,079 tokens in common, calculated as the intersection of their vocabularies. In contrast, MPT, developed by MosaicML, employs a significantly larger tokenizer vocabulary comprising 50,277 tokens, among which only 8,993 tokens overlap with LLaMA-2 and 9,761 with OpenLLaMA. These distinctions in vocabulary size and token overlap reflect the varying tokenization strategies adopted by different model developers and underscore the implications for compatibility and performance across different language models.

**Training Pipeline of PTA-LLM**    The training procedure consists of two stages: *Probabilistic Token Alignment (Offline Phase)* and *Supervised Training with an Alignment Objective*.

*Stage 1: Probabilistic Token Alignment (Offline Phase)*. In this stage, we compute a fused alignment matrix between all pairs of source and target models using probabilistic token alignment. The process comprises the following steps:

1. *Preprocessing:* Segment the MiniPile dataset by splitting long texts into shorter segments for manageable processing.

2. *Model Inference:* Perform inference on the segmented texts using both source and target models to extract per-step logits and token indices.

3. *Optimal Transport Alignment:* Apply optimal transport to align the per-step logits and token indices, thereby constructing a fused alignment matrix across source-target model pairs.

*Stage 2: Supervised Training with Alignment Objective.* In the second stage, we train the target model using a joint objective that combines the fused alignment matrix obtained from Stage 1 and the one-hot label matrix derived from the MiniPile dataset. This joint supervision encourages the model to learn both from the aligned token distributions of the source models and the ground-truth labels, facilitating effective knowledge transfer while preserving task-specific accuracy.

**Fusion Pipeline of PTA-LLM**  PTA-LLM requires only a single target model and supports any number of source models (at least one), regardless of their architecture or tokenizer. A key characteristic of this framework is that the target model undergoes end-to-end training, where its training objective is defined by a combination of the causal language modeling loss ($\mathcal{L}_{\text{CLM}}$) and the fusion loss ($\mathcal{L}_{\text{fusion}}$). In essence, while the final fused model retains the architecture of the target model, its parameters are enriched by integrating knowledge from the source models through the fusion process.

To illustrate the fusion process, we use our experimental setup as an example and extend it to a scenario involving an arbitrary number of source models. For fair comparison, we designate LLaMA-2 as the target model, with MPT and OpenLLaMA-2 serving as source models. Note that different target model choices may result in varying base performance, as discussed in prior works such as FUSELLM and FUSECHAT.

Prior to fusion, we perform a forward pass through LLaMA-2, OpenLLaMA-2, and MPT to obtain their probabilistic distribution matrices, denoted as $P_t$, $P_{s_1}$, and $P_{s_2}$, respectively. If additional source models (*e.g.*, DeepSeek or Qwen) are included, we similarly forward them to obtain their respective probabilistic matrices $P_{s_n}$ for the $n$-th source model.

The fusion process proceeds in a recursive manner as follows:

- **Fusion Stage 1:** Align $P_t$ and $P_{s_1}$ (OpenLLaMA-2) using the method described in Section 3.2, resulting in a fused matrix $P_{t,s_1}$.
- **Fusion Stage 2:** Align $P_{t,s_1}$ and $P_{s_2}$ (MPT), resulting in the updated matrix $P_{t,s_1,s_2}$.
- **Fusion Stage 3:** Align $P_{t,s_1,s_2}$ with $P_{s_3}$ (third source model), producing $P_{t,s_1,s_2,s_3}$.
- **Fusion Stage $n$:** Continue recursively, aligning $P_{t,s_1,\ldots,s_{n-1}}$ with $P_{s_n}$ to obtain the final fused matrix $P_{t,s_1,\ldots,s_n}$.

As demonstrated, our fusion strategy avoids the need for pairwise alignment between every target–source model pair, leading to linear complexity with respect to the number of source models. This efficient and scalable design makes the framework suitable for incorporating large collections of heterogeneous models.

## S3   Details of Dataset

• The Grade School Math [32], proposed by OpenAI, comprises a wide variety of conceptually simple grade school-level word problems and serves as a benchmark to assess the shortcomings of language models in handling multi-step mathematical *reasoning*. We evaluate it using the accuracy (8 shot) under the lm-evaluation-harness framework [33].

• Big-Bench Hard (BBH) [34] is a benchmark to evaluate the general *reasoning* ability of LLMs, containing 23 multiple-choice tasks and 4 free-form generation tasks from the Big-Bench [35]. We evaluate it using the EM accuracy based on few-shot chain-of-thought (CoT) prompts under the open-instruct framework following [11].

• MultiPL-E (ME) [36] is a multilingual programming benchmark to assess the *commonsense* ability of LLMs, consisting of 18 different programming languages with 17 parallel datasets translated from the Python benchmark [37]. We evaluate it using pass@1 [37] based on 20 generated samples

for each question in 10 popular programming languages under the bigcode-evaluation-hardness framework [38, 11].

• Measuring Massive Multitask Language Understanding (MMLU) [39] is a massive multitask test consisting of multiple-choice questions from various branches of knowledge to assess the *commonsense* ability of LLMs, including 17 sub categories (*i.e.*, US history, computer science and law) that people must study to learn. We evaluate it using the classification accuracy under the open-instruct framework.

• ToxiGen [40] is a large-scale machine-generated dataset for adversarial and implicit hate speech detection used to evaluate the *safty* ability of LLMs, which contains implicitly toxic and benign sentences mentioning 14 minority groups. We evaluate it using the non-toxicity rate (*i.e.*, 1 - reported toxicity rate) under the open-instruct framework.

• TyDi QA [41] is a benchmark for information-seeking question answering in typologically diverse languages to asses the *multilingual* ability of LLMs. It covers 9 different languages including korean, arabic, indonesian, *etc*. We evaluate it using the EM accuracy under the open-instruct framework.

## S4    Notations

Table S1: The main notations used in PTA-LLM.

| Notation | Representation |
|---|---|
| $\mathcal{C}$ | A training corpus consisting of a collection of text sequences $t$ |
| $t$ | Text sequences with a maximum length of 2048 tokens |
| $\mathcal{P}_s$ | The probabilistic distribution matrices for the source model, consisting of $L$ tokens $\mathcal{A}$ |
| $V_s$ | The vocabulary sizes of the source model tokenizer |
| $L$ | The sequence length in $\mathcal{P}_s$ |
| $\mathcal{A}$ | Tokens in $\mathcal{P}_s$ have logits over a vocabulary of size $V_s$ |
| $\mathcal{P}_t$ | The probabilistic distribution matrices for the target model, consisting of $N$ tokens $\mathcal{B}$ |
| $V_t$ | The vocabulary sizes of the source model tokenizer |
| $N$ | The sequence length in $\mathcal{P}_t$ |
| $\mathcal{B}$ | Tokens in $\mathcal{P}_t$ have logits over a vocabulary of size $V_t$ |
| $\mathcal{P}_f$ | The fused probabilistic distribution matrices for model fusion, consisting of $N$ tokens $\hat{\mathcal{B}}$ |
| $\hat{\mathcal{B}}$ | Fused tokens in $\mathcal{P}_f$ have logits over a vocabulary of size $V_t$ |
| $\mathcal{L}$ | The overall training objective of PTA-LLM |
| $\mathcal{L}_{\text{CLM}}$ | The training objective of casual language modeling |
| $\mathcal{L}_{\text{Fusion}}$ | The training objective of kownledge fusion |
| $\mathcal{L}_{\text{Fusion}}$ | The training objective of kownledge fusion |
| $\lambda$ | A hyperparameter controlling the trade-off between the $\mathcal{L}_{\text{CLM}}$ and the $\mathcal{L}_{\text{Fusion}}$ |
| $\mathbf{Q}_t$ | Target model's predictions during the training |
| $\mathbf{O}_t$ | One-hot label matrix in $t$ |
| $\mathcal{T}$ | The initial $n \times m$ optimal transport plan matrix |
| $\hat{\mathcal{T}}$ | The final $n \times m$ optimal transport plan matrix of non-negative entries $\mathcal{G}_{nm}$ |
| $\mathcal{G}_{nm}$ | The amount of logit probability transported from the $n$-th source indice to the $m$-th target indice. |
| $C$ | An $n \times m$ transport cost matrix of non-negative entries $c_{xy}$ |
| $c_{xy}$ | The alignment cost between source indice and target indice |
| $R_x$ | The indices corresponding to the largest values in the $x$-th row of $\hat{\mathcal{T}}$ |

# S5  Details of Probabilistic Token Alignment

Our training procedures are implemented based on the publicly available code from [11], with modifications made specifically to the token alignment module. For better understanding, the following is a concise pseudo code of §3.2. Specifically, we perform optimal transport on logits after applying the softmax function to reduce the impact of extreme values (*e.g.*, extreme large, small, or negative values) that could otherwise distort the transport cost. Importantly, conducting transport in logit space differs fundamentally from transporting mass in probability space due to the distinct normalization terms associated with the source and target spaces. We plan to investigate it further in the future. Our full implementation shall be publicly released upon paper acceptance.

---

**Algorithm 2** Probabilistic Token Alignment

---

**Require:** Tokenizer, input IDs, per step logits, per step indices from both source and target Model.
 1: Convert input IDs to token sequence.
 2: Use Dynamic Programming in 1 to obtain token pairing between two token sequences.
 3: **for** each token pairing **do**
 4:    **if** it is a one-to-one token pairing **then**
 5:       use the sinkhorn algorithim in S2 under the optimal transport framework, considering per step logits and indices from source and target token
 6:    **else**
 7:       use the one-hot logits
 8:    **end if**
 9: **end for**
10: **return**  Aligned matrix

---

# S6  Visualization of Token Alignment

In this section, we present more details and results of visualization of token alignment to support our findings in §4.4. All samples are the token alignment of target model (*i.e.*, Llama) and source model (*i.e.*, MPT).

In Fig.S1, we can first observe a significant center shift in FUSELLM while our method maintain its overall distribution, showing consistency with our paper.

In Fig.S2 and Fig.S3, we present more visualization inspection results for FUSELLM and PTA-LLM using Isomap [29] and PCA [30]. Overall, we present additional visual evidence to support the notion that the probabilistic token alignment generate a more compact and coherent representation.

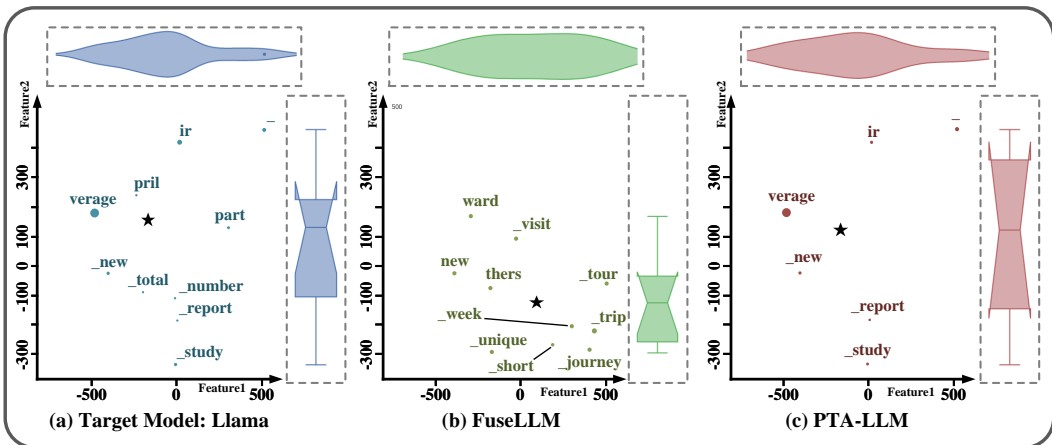

Figure S1: **Sample A.** 2D visualization results of target tokens and fused tokens.

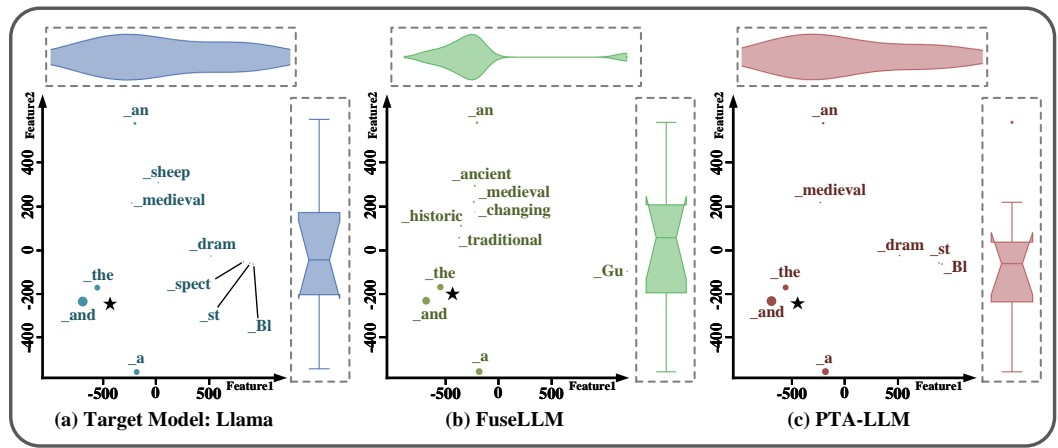

Figure S2: **Sample B.** 2D visualization results of target tokens and fused tokens.

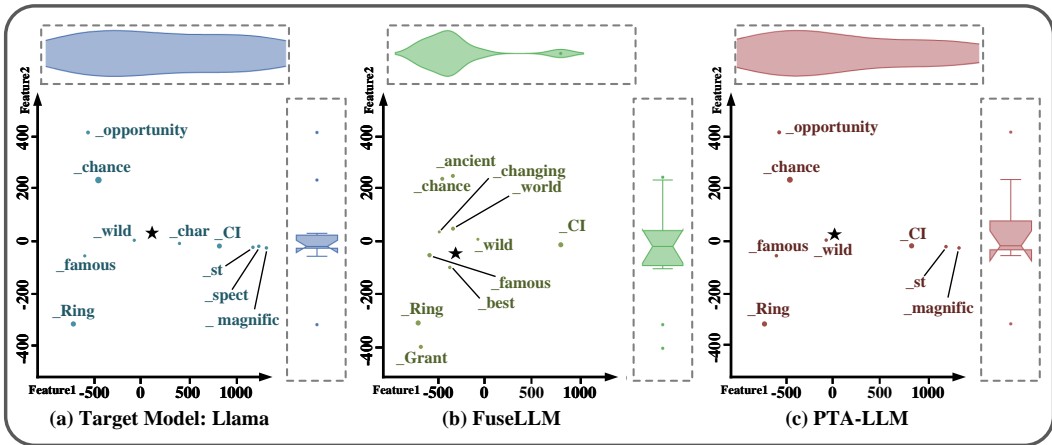

Figure S3: **Sample C.** 2D visualization results of target tokens and fused tokens.

## S7  Pilot Study

### S7.1  Heterogeneous Fusion

We also acknowledge related work [19], which explores the fusion of Mixtral, InternLM2, and Open-Chat, demonstrating consistent performance improvements under the knowledge fusion paradigm. Motivated by these findings, we conduct a preliminary experiment on MiniPile to extend our framework for fusing additional LLMs. Our experimental setup would further benefit from incorporating models with greater architectural diversity. As shown in the Tab. S2 and S3, both "PTA-LLM + Qwen2.5" and "PTA-LLM + Mistral" outperform Qwen2.5 and Mistral, respectively, highlighting the effectiveness of our approach.

Table S2: The baseline models used in this pilot study include more heterogeneous models such as Mixtral [42] and Qwen [43].

| Benchmark | OpenLLaMA | MPT | Llama-2 | Mistral | Qwen2.5 |
|-----------|-----------|-----|---------|---------|---------|
| *Baseline Performance* | | | | | |
| MMLU | 42.11 | 27.84 | 46.94 | 59.15 | 71.80 |
| Hellaswag | 74.52 | 76.35 | 75.99 | 80.39 | 78.89 |

Table S3: The fusion results using the baseline model in Tab. S2 under multiple choice task.

| Benchmark | FuseLLM | PTA-LLM (ours) | PTA-LLM + Mistral | PTA-LLM + Qwen2.5 |
|---|---|---|---|---|
| *Fusion Performance* | | | | |
| MMLU | 48.77 | 49.38 | 60.34 | 72.66 |
| Hellaswag | 78.23 | 79.74 | 81.52 | 80.13 |

For generative tasks, we have already reported results on MultiPL-E in the original paper. To further compare our approach with the Mistral and Qwen models, we conducted additional experiments on HumanEval, using "MiniPile+Github" for training. The results presented below in Tab. S4, together with our original findings on MultiPL-E, further validate the effectiveness of our method for generative tasks.

Table S4: The fusion results using the baseline model in Tab. S2 under generative task.

| Benchmark | Mistral | Qwen2.5 | PTA-LLM + Mistral | PTA-LLM + Qwen2.5 |
|---|---|---|---|---|
| *Fusion Performance* | | | | |
| HumanEval | 30.50 | 57.90 | 32.64 | 59.71 |

## S7.2  Training Burden on Knowledge Fusion

To further validate the effectiveness of konwledge fusion, we conducted an additional experiment comparing two distinct settings. **Setting A**: Train on a subset of MiniPile (100K examples, randomly sampled as 10% of the original dataset) using the same configuration as in the paper. The total training cost, including obtaining probability distributions from all teacher models ( 3.2 GPU hours) and end-to-end training ( 20 GPU hours), amounts to 23.2 GPU hours. **Setting B**: Allocate the same computational budget to train on a larger dataset. Specifically, we train Llama-2 CLM on a larger MiniPile subset (116K examples, corresponding to 11.6% of the original dataset), ensuring the total training cost remains 23.2 GPU hours, matching Setting A.

The results, summarized in the Tab. S5, align with our previous findings, reinforcing the advantages of the knowledge fusion paradigm under equivalent resource constraints.

Table S5: The performance under the same computation cost.

| Benchmark | Setting A (Our method) | Setting B (CLM with more data) |
|---|---|---|
| MMLU | 47.42 | 46.33 |

## S7.3  Training Time

Since the end-to-end training time is primarily determined by the dataset size, we conducted an additional study using subsets of MiniPile in Tab. S6, which consists of 1M training samples. Specifically, we randomly sampled the original dataset to create training subsets. The results reveal a linear relationship between dataset size and training time. All experiments were performed on 8 NVIDIA A100-80GB GPUs. Notably, we further evaluate the model under different settings. PTA-LLM achieves MMLU scores of 49.38, 48.55, and 47.42 when trained on MiniPile datasets of 1M (100%), 500K (50%), and 100K (10%) samples, respectively. The results show that larger dataset sizes yield modest performance improvements. Notably, even with only 10% of the data, PTA-LLM attains a score of 47.42 on MMLU, which still surpasses the traditional CLM baseline of 47.65.

Table S6: The fusion results using the baseline model in Tab. S2 under generative task.

| | MiniPile 1M (100%) | MiniPile 500K (50%) | MiniPile 100K (10%) | MiniPile 10K (1%) |
|---|---|---|---|---|
| Training Time | 26.0h | 12.9 h | 2.4h | 50.3h |

# S8 Per-task Results on Different Benchmarks

For the training acceleration, we leverage Deepseepd [44] and FlashAttention [28]. More specifically, we optimize our model using the AdamW optimizer, with hyperparameters set to $\beta_1 = 0.9$ and $\beta_2 = 0.95$, applying gradient clipping at 1.0 and a weight decay of 0.05. The learning rate follows a cosine schedule, peaking at $1 \times 10^{-5}$, with a warmup ratio of 0.008.

To provide comprehensive results from the paper, we report the average per-benchmark results on The Grade School Math, Big-Bench Hard, MultiPL-E, Measuring Massive Multitask Language Understandin, ToxiGen and TyDi QA respectively (see Table 2). We note that the results of all methods in Table 2 have been rerun with the same configuration on our own machine (*i.e.*, 8 NVIDIA H100-80GB GPUs) and may therefore exhibit slight variations compared to other reports. Furthermore, we report per-task results (78 tasks) here in Table S7 for better clarification.

Our results are statistically significant with respect to all baselines on each benchmark (all p-value < 0.005). Furthermore, we rerun the same hyperparameter settings three times and computed standard deviation error bars for BBH, MMLU and ME benchmark.

Table S7: **PTA-LLM** per-task results on six various benchmark.

| Task | PTA-LLM | Task | PTA-LLM |
|---|---|---|---|
| *Grade School Math* | | *MMLU std=0.05* | |
| Grade School Math | 14.71 | Math | 31.30 |
| *Big-Bench Hard (BBH) std=0.04* | | Health | 50.91 |
| Boolean Expressions | 68.40 | Physics | 37.66 |
| Causal Judgement | 50.80 | Business | 62.93 |
| Date Understanding | 58.80 | Biology | 53.96 |
| Disambiguation QA | 48.00 | Chemistry | 36.96 |
| Dyck Languages | 3.20 | Computer Science | 45.39 |
| Formal Fallacies | 46.00 | Economics | 42.59 |
| Geometric Shapes | 26.80 | Engineering | 51.72 |
| Hyperbaton | 64.00 | Philosophy | 41.40 |
| Logical Deduction (3 objects) | 59.60 | Other | 57.94 |
| Logical Deduction (5 objects) | 36.00 | History | 59.57 |
| Logical Deduction (7 objects) | 26.40 | Geography | 53.03 |
| Movie Recommendation | 69.20 | Politics | 58.33 |
| Multistep Arithmetic Two | 4.00 | Psychology | 55.49 |
| Navigate | 60.00 | Culture | 61.45 |
| Object Counting | 56.40 | Law | 38.80 |
| Penguins in a Table | 36.30 | Avg. 17 Tasks | 49.38 |
| Reasoning about Colored Objects | 52.40 | *ToxiGen* | |
| Ruin Names | 30.00 | Black | 12.60 |
| Salient Translation Error Detection | 26.40 | Mexican | 8.00 |
| Snarks | 47.19 | LGBTQ | 24.00 |
| Sports Understanding | 91.60 | Jewish | 25.20 |
| Temporal Sequences | 15.20 | Women | 37.20 |
| Tracking Shuffled Objects (3 objects) | 30.40 | Middle East | 11.00 |
| Tracking Shuffled Objects (5 objects) | 17.20 | Muslim | 12.60 |
| Tracking Shuffled Objects (7 objects) | 14.00 | Trans | 22.40 |
| Web of Lies | 64.80 | Asian | 36.40 |
| Word Sorting | 6.00 | Physical Disability | 17.80 |
| Avg. 27 Tasks | 41.08 | Latino | 16.60 |
| *MultiPL-E (ME) std=0.05* | | Native American | 6.20 |
| C++ | 9.75 | Chinese | 23.20 |
| Go | 64.51 | Mental Disability | 11.20 |
| Java | 9.88 | Avg. 14 Tasks | 18.89 |
| JavaScript | 13.85 | *TyDi QA* | |
| PHP | 9.10 | Arabic | 9.55 |
| Python | 13.87 | Bengali | 21.24 |
| R | 5.75 | English | 55.23 |
| Ruby | 11.58 | Finnish | 43.22 |
| Rust | 7.24 | Indonesian | 46.02 |
| TypeScript | 13.26 | Korean | 55.80 |
| Avg. 10 Tasks | 15.88 | Russian | 33.74 |
| | | Swahili | 41.68 |
| | | Telugu | 0.15 |
| | | Avg. 9 Tasks | 34.07 |

## S8.1 Additional Comparison Results

We provide additional comparison results for cases where FuseLLM performs well while PTA-LLM performs poorly.

Table S8: Comparison of FuseLLM and PTA-LLM on BBH subtasks (EM accuracy).

| Task | LLaMA-2 | FuseLLM | PTA-LLM |
|---|---|---|---|
| Multistep Arithmetic Two | 0.80 | 4.80 | 4.00 |
| Temporal Sequences | 12.80 | 16.40 | 15.20 |

As shown in the Table above, PTA-LLM exhibits slight performance degradation in two BBH subtasks, both of which evaluate EM accuracy on various reasoning scenarios. PTA-LLM applies OT to balance the probability mass in each row, producing a soft assignment matrix that yields more compact and consistent token representations. This smoothing flattens peaks for high-confidence tokens. In exact-match BBH tasks, spreading probability to distractor tokens can shift the argmax and reduce accuracy. PTA-LLM is penalized for minor surface deviations, while FuseLLM's hard alignment preserves high-confidence tokens, yielding more frequent matches with the official answers. This explains why PTA-LLM achieves the smallest accuracy improvement under BBH in Table 2.

## S9 Case Studies

In Table S9, Table S10, and Table S11, we present case studies in difficult tasks (see the overall peformance results in Table 3) to demonstrate the stability of PTA-LLM.

Table S9: Case studies on the Causal Judgement task.

---

**BBH; Causal Judgement;**

**Question:** How would a typical person answer each of the following questions about causation?:
A machine is set up in such a way that it will short circuit if both the black wire and the red wire touch the battery at the same time. The machine will not short circuit if just one of these wires touches the battery. The black wire is designated as the one that is supposed to touch the battery, while the red wire is supposed to remain in some other part of the machine. One day, the black wire and the red wire both end up touching the battery at the same time. There is a short circuit. Did the black wire cause the short circuit?
Options: Yes / No

**Target:** No

**FUSELLM:** Let's think step by step.Here in this question, we are told that "The black wire is designated as the one that is supposed to touch the battery, while the red wire is supposed to remain in some other part of the machine." A typical person would therefore assume that the black wire was supposed to touch the battery and that the red wire was supposed to remain in some other part of the machine. Because the black wire and the red wire both ended up touching the battery at the same time, a typical person would therefore conclude that the black wire caused the short circuit. So the answer is Yes.

**PTA-LLM:** Let's think step by step.Here in this question, we are told that "The black wire is designated as the one that is supposed to touch the battery, while the red wire is supposed to remain in some other part of the machine." A typical person would assume that the black wire is supposed to touch the battery, while the red wire is supposed to remain in some other part of the machine. A typical person would therefore conclude that the black wire did not cause the short circuit. So the answer is No.

---

Table S10: Case studies on the Tracking Shuffled Objects (7 objects) task.

| BBH; Tracking Shuffled Objects (7 objects); |
| --- |
| **Question:** Alice, Bob, Claire, Dave, Eve, Fred, and Gertrude are dancers at a square dance. At the start of a song, they each have a partner: Alice is dancing with Lola, Bob is dancing with Rodrigo, Claire is dancing with Jamie, Dave is dancing with Izzi, Eve is dancing with Sam, Fred is dancing with Karl, and Gertrude is dancing with Helga. Throughout the song, the dancers often trade partners. First, Alice and Claire switch partners. Then, Bob and Fred switch partners. Then, Claire and Dave switch partners. Then, Alice and Fred switch partners. Then, Claire and Alice switch partners. Then, Bob and Eve switch partners. Finally, Eve and Gertrude switch partners. At the end of the dance, Eve is dancing with
Options: (A) Lola (B) Rodrigo (C) Jamie (D) Izzi (E) Sam (F) Karl (G) Helga |
| **Target:** (G) |
| **FUSELLM:** Let's think step by step. (0) At the start: Alice: Lola, Bob: Rodrigo, Claire: Jamie, Dave: Izzi, Eve: Sam, Fred: Karl, Gertrude: Helga. (1) Alice and Claire switch partners: Alice: Rodrigo, Claire: Lola, Dave: Izzi, Eve: Sam, Fred: Karl, Gertrude: Helga. (2) Bob and Fred switch partners: Alice: Rodrigo, Claire: Lola, Dave: Izzi, Eve: Sam, Fred: Karl, Gertrude: Helga. (3) Claire and Dave switch partners: Alice: Rodrigo, Claire: Lola, Bob: Izzi, Eve: Sam, Fred: Karl, Gertrude: Helga. (4) Alice and Fred switch partners: Alice: Izzi, Claire: Lola, Bob: Rodrigo, Dave: Sam, Eve: Karl, Gertrude: Helga. (5) Claire and Alice switch partners: Alice: Izzi, Claire: Lola, Bob: Rodrigo, Dave: Sam, Eve: Karl, Gertrude: Helga. (6) Bob and Eve switch partners: Alice: Izzi, Claire: Lola, Bob: Rodrigo, Dave: Sam, Eve: Karl, Gertrude: Helga. (7) Eve and Gertrude switch partners: Alice: Izzi, Claire: Lola, Bob: Rodrigo, Dave: Sam, Eve: Karl, Gertrude: Helga. At the end of the dance, Eve is dancing with Karl. So the answer is (F). |
| **PTA-LLM:** Let's think step by step. (0) At the start: Alice: Lola, Bob: Rodrigo, Claire: Jamie, Dave: Izzi, Eve: Sam, Fred: Karl, Gertrude: Helga. (1) Alice and Claire switch partners: Alice: Jamie, Claire: Lola, Dave: Izzi, Eve: Sam, Fred: Karl, Gertrude: Helga. (2) Bob and Fred switch partners: Alice: Izzi, Claire: Lola, Bob: Rodrigo, Dave: Jamie, Eve: Sam, Fred: Karl, Gertrude: Helga. (3) Claire and Dave switch partners: Alice: Izzi, Claire: Rodrigo, Bob: Jamie, Dave: Lola, Eve: Sam, Fred: Karl, Gertrude: Helga. (4) Alice and Fred switch partners: Alice: Sam, Claire: Rodrigo, Bob: Izzi, Dave: Lola, Eve: Jamie, Fred: Karl, Gertrude: Helga. (5) Claire and Alice switch partners: Alice: Sam, Claire: Izzi, Bob: Rodrigo, Dave: Lola, Eve: Jamie, Fred: Karl, Gertrude: Helga. (6) Bob and Eve switch partners: Alice: Sam, Claire: Izzi, Bob: Rodrigo, Dave: Lola, Eve: Jamie, Fred: Karl, Gertrude: Helga. (7) Eve and Gertrude switch partners: Alice: Sam, Claire: Izzi, Bob: Rodrigo, Dave: Lola, Eve: Jamie, Fred: Karl, Gertrude: Helga. At the end of the dance, Eve is dancing with Helga. So the answer is (G). |

As shown in Table S9, PTA-LLM's response is correct because it accurately identifies the key element of causation in the scenario. The question specifies that the black wire is expected to touch the battery as part of the machine's normal setup, while the red wire is not supposed to do so. When the short circuit occurs, the black wire's action is consistent with its intended role and does not deviate from normal functioning. On the other hand, the red wire's unexpected contact with the battery introduces the condition necessary for the short circuit. PTA-LLM correctly reasons that the red wire's abnormal behavior is the true cause of the short circuit, aligning with how a typical person would perceive causation. In contrast, FuseLLM overlooks the normalcy of the black wire's role and incorrectly attributes causation to it, simply because both wires were involved. This makes PTA-LLM's reasoning more logical and consistent with the principles of causation.

As shown in Table S10, tracking shuffled objects task with seven objects is a particularly challenging scenario requiring accurate tracking of the corresponding dancers among seven individuals as they switch partners many times.In this context, FuseLLM fails to track the objective during the fourth partner switch, whereas PTA-LLM successfully tracks the corresponding dancers throughout. This superior performance is likely attributable to PTA-LLM's probabilistic token alignment mechanism, which effectively transforms logits into the correct objective rather than merely replicating the original logits in the FuseLLM approach.

As shown in Table S11, PTA-LLM correctly identifies the SVG path as forming a hexagon, recognizing 7 commands: one "M" to start and six "L" commands creating a closed six-sided polygon. FuseLLM miscounts the commands, identifying only 5, and incorrectly concludes the shape is a pentagon. PTA-LLM's accurate command count and shape identification make its reasoning correct.

Table S11: Case studies on the Geometric Shape task.

| |
|---|
| **BBH; Geometric Shape;** |
| **Question:** This SVG path element <path d="M 60.00,77.00 L 75.00,55.00 L 6.00,60.00 L 22.00,78.00 L 3.00,96.00 L 93.00,90.00 L 60.00,77.00"/> draws a
Options: (A) circle (B) heptagon (C) hexagon (D) kite (E) line (F) octagon (G) pentagon (H) rectangle (I) sector (J) triangle" |
| **Target:** (C) |
| **FUSELLM:** Let's think step by step. This SVG path element contains "M" and "L" commands. M takes two parameters (x,y) and moves the current point to the coordinates (x,y). L takes two parameters (x,y) and draws a line from the previous coordinate to the new coordinate (x,y). This path can be decomposed into 6 separate commands. (1) M 60.00,77.00: Move the current point to 60.00,77.00. (2) L 75.00,55.00: Create a line from 60.00,77.00 to 75.00,55.00. (3) L 6.00,60.00: Create a line from 75.00,55.00 to 6.00,60.00. (4) L 22.00,78.00: Create a line from 6.00,60.00 to 22.00,78.00. (5) L 3.00,96.00: Create a line from 22.00,78.00 to 3.00,96.00. (6) L 93.00,90.00: Create a line from 3.00,96.00 to 93.00,90.00. This SVG path starts at point 60.00,77.00, creates five consecutive and touching lines, and then returns back its starting point, thereby creating a five-sided shape. It does not have any curves or arches. "pentagon" is the only five-sided polygon on the list. So the answer is (G). |
| **PTA-LLM:** Let's think step by step. This SVG path element contains "M" and "L" commands. M takes two parameters (x,y) and moves the current point to the coordinates (x,y). L takes two parameters (x,y) and draws a line from the previous coordinate to the new coordinate (x,y). This path can be decomposed into 7 separate commands. (1) M 60.00,77.00: Move the current point to 60.00,77.00. (2) L 75.00,55.00: Create a line from 60.00,77.00 to 75.00,55.00. (3) L 6.00,60.00: Create a line from 75.00,55.00 to 6.00,60.00. (4) L 22.00,78.00: Create a line from 6.00,60.00 to 22.00,78.00. (5) L 3.00,96.00: Create a line from 22.00,78.00 to 3.00,96.00. (6) L 93.00,90.00: Create a line from 3.00,96.00 to 93.00,90.00. (7) L 60.00,77.00: Create a line from 93.00,90.00 to 60.00,77.00. This SVG path starts at point 60.00,77.00, creates six consecutive and touching lines, and then returns back its starting point, thereby creating a six-sided shape. It does not have any curves or arches. "hexagon" is the only six-sided object on the list. So the answer is (C). |

## S10 Ablative Studies

Table S12: Ablative studies of optimal transport convergence threshold

| Choice | BBH | ME | MMLU |
|---|---|---|---|
| *Optimal Transport Convergence Threshold* | | | |
| 1e-3 | 39.44 | 15.10 | 48.23 |
| 1e-4 | 40.54 | **15.88** | 48.99 |
| 5e-5 | 40.91 | 15.85 | 49.32 |
| 1e-5 | **41.08** | 15.82 | **49.38** |
| 1e-6 | 41.04 | 15.78 | 49.35 |
| 1e-7 | 41.05 | 15.80 | 49.33 |

As shown in Table S12, the findings on the optimal transport convergence threshold align with our motivation. Specifically, a lower threshold preference suggests that stricter constraints may generate a more coherent fusion, leading to greater performance gains. We also observe that performance stabilizes when the threshold drops below 1e-5, suggesting that the transported cost is fully optimized and remains unchanged.

Table S13: Ablative studies of token alignment window size

| Choice | BBH | ME | MMLU |
|---|---|---|---|
| *Token Alignment Window Size* | | | |
| 10 | **41.08** | **15.88** | 48.99 |
| 7 | 40.99 | 15.73 | 49.00 |
| 5 | 40.68 | 15.61 | **49.38** |
| 3 | 39.64 | 15.08 | 47.11 |

The motivation for using an alignment window is that, without constraints on $m$ and $n$, the transport space between the logits of Model A and Model B would be the product of their vocabulary sizes (*e.g.*, MPT has 50,277 tokens while LLaMA has 32,000). This is computationally inefficient, as it requires storing over 30k logits per token instead of just the top 10, and unnecessary because the logit distribution follows a long-tail pattern where the top-k logits capture most of the probability mass. Our analysis of token alignment window sizes, ranging from 10 to 3, shows that a small window is sufficient for effective knowledge fusion. As shown in Table S13, larger windows generally improve

BBH and ME performance, with size 10 performing best on both. MMLU peaks at size 5, while very small windows (*e.g.*, 3) reduce performance across tasks. Overall, window sizes between 5–10 offer consistently strong results.

Table S14: Ablative studies of combination weight

| Choice | BBH | ME | MMLU |
|--------|-----|-----|------|
| *Combination Weight* | | | |
| 0.90 | 40.39 | 15.72 | 48.93 |
| 0.85 | 41.00 | **15.91** | 49.09 |
| 0.80 | **41.08** | 15.88 | **49.38** |
| 0.75 | 39.78 | 15.65 | 47.29 |
| 0.70 | 38.11 | 14.27 | 46.08 |
| 0.60 | 37.20 | 14.09 | 45.96 |

As shown in Table S14, it further reveals that the observed "higher performance when the weight is smaller" pertains specifically to the comparison between 0.8 and 0.9. However, if the weight is reduced further, the model overemphasizes the fused matrix and pays less attention to the original CLM modeling. Consequently, we selected 0.8 for all experiments, as it consistently achieves the best performance.

## S11    Future Work and Discussions

**Limitation.** A limitation of our approach is that the Sinkhorn-Knopp algorithm runs in $\widetilde{O}(n^2/\epsilon^3)$ time, which reduces the token alignment efficiency. Despite the observation that in practice only 3 Sinkhorn loops per training iteration are often sufficient for model representation, which amounts to ~13.75% aligning delay on MiniPile compared with FUSELLM. It would be interesting to investigate further lower complexity (*i.e.*, greenkhorn [45]) algorithim to compute the optimal transport.

**Future Work.** Despite PTA-LLM systemic generality (see §4.2) and robustness (see §4.3), it also comes with new challenges and unveils some intriguing questions. For instance, the overall pipeline is divided into two stages: alignment and fusion training. This naturally raises an important question from a paradigm perspective: Can we design an end-to-end fusion pipeline that dynamically controls token alignment, thereby enabling more comprehensive capability learning? Introducing a new loss design (*i.e.*, universal logit distillation loss [46]) within the fusion training to deal with the misalignment problem in different tokenizers might enhance pipeline efficiency and facilitate additional performance improvements. Another essential future direction deserving of further investigation is its further effectiveness exploration in other NLP fields since aligning sequences generated by different tokenizers is a generic problem of contemporary NLP. In §4.4, we demonstrate through visualization studies that probabilistic token alignment yield a more coherent fused representation. Consequently, the applicability of this integration to other alignment methods requires further investigation.

In this paper, we do not fully explore the potential of knowledge fusion, as comprehensive experiments on heterogeneous models remain outside the scope of our study. However, related work [19] has investigated the fusion of models such as Mixtral [47], InternLM2 [48], and OpenChat [49], demonstrating consistent performance improvements within the knowledge fusion paradigm. We plan to explore it further in the future. Inspired by [50, 51], PTA-LLM could also incorporate learnable soft prompts, enabling task-specific adaptation without retraining the entire alignment matrix.

Besides the directions mentioned earlier, we identify several additional promising avenues for exploration. First, an end-to-end fusion pipeline could streamline the process and reduce the reliance on CPU resources by eliminating the need for a two-stage approach (alignment followed by training). This could be facilitated by leveraging innovative loss functions to enable dynamic adjustments. Second, the exploration of N-1 and 1-N mapping strategies offers enhanced flexibility. While this paper focuses on 1-1 mapping due to constraints imposed by traditional optimal transport frameworks, future work could explore beyond these limitations. Lastly, multilingual alignment, such as aligning Chinese and English tokens, holds the potential to broaden applicability, as current research predominantly focuses on English token alignment.

**Discussion.** Two potential factors may explain why the knowledge fusion objective outperforms the traditional CLM approach: First, the CLM objective employs one-hot vectors as the golden labels, which fails to capture the nuanced information each token might convey. This approach provides the

same penalty for completely incorrect predictions as for predictions that select an incorrect token but retain semantically relevant context. In other words, the CLM objective does not reward predictions that are "almost correct," which limits its capacity to encourage fine-grained improvements. Second, the fusion objective incorporates representations from diverse source models through distillation, enabling it to capitalize on the complementary strengths of each model. It provides more fine-grained context information for alignment.

Regarding the performance, our performance improvements are constrained by the suboptimal performance of certain source LLMs relative to the target LLM on specific tasks, which inevitably impacts the quality of the fusion results. We also observe that the performance improvement could be significantly enhanced by increasing the size of the continued training datasets. Notably, the original MiniPile [25] comprises only 8% coding-related data. By incorporating the GitHub datasets from the Pile [52] in our priliminary experiments, it is possible to achieve greater performance gains, particularly in coding-related downstream tasks.

Regarding the motivations behind the Equation 5. To achieve a coherent fused metric as described in our introduction, we aim to keep the logit distribution more compact and consistent with the target distribution. Due to the constraints in optimal transport, we discretely select the maximum value in each sub-transport, which helps maintain both compactness and consistency. We are also interested in whether broader transport (*e.g.*, choosing top-k instead of top-1) could further improve fusion, and we plan to investigate this in future work.

Regarding the use of edit distance. Our motivation for using minimum edit distance as the evaluation metric comes from the scale of our 1M-sample training corpus, where even a small increase in complexity could create a significant computational burden for token alignment. A semantic similarity-based token-level metric could embed more comprehensive and reasonable information for transportation. We are very interested in this direction and plan to investigate it further, exploring more efficient semantic similarity approaches that are well-suited to our setting.

**Potential Risks.** Consider the tuning process of LLM, which has potential risks for energy usage. Finetuning requires significant computational power, leading to high energy use and increased environmental impact.

**Asset License and Consent.** The OpenLLaMA [4], bigcode-evaluation-harness [38] and MPT [5] are licensed under Apache-2.0; Llama-2 7b [3] is licensed under Llama 2 Community License Agreement; lm-evaluation-harness [33] is licensed under MIT;

**Artifact Consistent With Intended Use.** Our work ensures that the use of existing artifacts aligns with their intended purpose when specified. For the artifacts we create, it remains compatible with the original access conditions. In particular, we ensure that derivatives of data accessed for research purposes are confined to research contexts.

**Social Impact.** This work introduces PTA-LLM, which demonstrates significant performance improvements over state-of-the-art baselines, as shown in Table 2. Our approach enhances model accuracy and is particularly beneficial for efficient training scenarios, such as on resource-constrained devices and rapid adaptation with limited computational overhead.

