# OpenReview forum: "Probabilistic Token Alignment for Large Language Model Fusion"
_NeurIPS.cc/2025/Conference — NeurIPS 2025 poster_

### Official Review · Reviewer_zYWa · 2025-06-24

**Clarity:** 2
**Significance:** 2
**Originality:** 3
**Rating:** 4
**Confidence:** 4

**Summary:**

This paper aims to fuse language models with different tokenizers together to form a stronger model. It is challenging because both the token sequence length and vocabulary size might vary. To address the token sequence length discrepancy, the authors employ a dynamic programming algorithm to compute the optimal mapping. To address the vocabulary discrepancy, the authors model the problem as an optimal transport question and solve it to get align the two distributions between two models. Experiment results confirm the proposed method lead to better model performance compared to previous model fusion and non-fused models.

**Questions:**

1. Is the recursion function defined in (1) exactly same as the recursion function defined in [1] (Eq (1) (2) (3))? Why it is always +C(B_k, A_j) for all three cases? I think it should be C(B_k) and C(A_j) for the first two lines, which means the cost of adding a new token.

2. When defining recursion function (1), do two sequences of tokens represent the same text? If so, then I think it is impossible that A2 matches B1 while A1 matches B2, correct? Because the order of tokens should follow the words in the sentences. But in Figure 2b(1), it illustrates mappings with perturbed orders. Why is it?

3. I think the process to compute token alignement and optimal transport is done on CPU and hard to be parallelized. I wonder its relative time cost compared to conventional pre-training.

4. Table 3 in Section 4.3 compares the performance of PTA-LLM and FuseLLM on seven tasks that FuseLLM perform worst. The authors claim PTA-LLM is more stable because it has better performance on these tasks. But the task selection is biased. The authors should report the tasks where PTA-LLM has the worst performance and compare the performance degradation with FuseLLM. I think the current setting is problematic.

5. I think Section 4.4 and Figure 3 are confusing. What is the figure trying to show?

6. Why in Eq (4) n=m=10?

[1] Fu, Yao, et al. "Specializing smaller language models towards multi-step reasoning." International Conference on Machine Learning. PMLR, 2023.

**Ethical Concerns:**

["NO or VERY MINOR ethics concerns only"]

**Final Justification:**

The authors addressed my concerns. So I maintain the positive score.

**Limitations:**

yes

**Quality:**

3

**Strengths And Weaknesses:**

S1. I think the proposed optimal transport method to align logits is reasonable and sound.

S2. The experiment results show consistend improvement on model performance.

W1. The dynamic token pairing technique is proposed by previous work, but the paper seems to claim it is original.

W2. Some experiment settings are potentially problematic.

For details of the weakness, see my questions below

---

> ### Author Rebuttal · Authors · 2025-07-31
>
> Dear reviewer zYWa,
>
> We sincerely appreciate your time and effort in reviewing our paper and providing valuable comments. We provide explanations to your questions point-by-point in the following.
>
> **Q1: Regarding Originality Claim of Dynamic Token Pairing.**
>
> Sorry for the confusion. Our token pairing is greatly inspired by [ref1] (line 88), which serves as a foundation for PTA‑LLM. We include its details in Section 3.2 to help readers better understand token pairing as the basis for our key contribution, Probabilistic Token Alignment through Optimal Transport, which is built upon previous token‑level pairing. We recognize that this presentation may cause confusion regarding proper credit. We sincerely appreciate prior work in establishing a reliable and efficient token pairing method and will revise the text to make this clearer.
>
> [ref1] Specializing smaller language models towards multi-step reasoning
>
> **Q2: Regarding Correctness of Recursion Function Definition in Equation (1).**
>
> Thank you for raising this point. Our recurrence in Eq. (1) is mathematically identical to Eqs. (1)–(3) in [1]; we simply present it in a more compact form:
>
> $f(k,j) = \min\lbrace f(k-1,j),\; f(k,j-1),\; f(k-1,j-1) \rbrace + C(B_k,A_j)$
>
> Our ultimate objective is to optimize $f(L, N)$f, whose minimum value is determined by the optimal path through the sum of subfunctions $ f(k, j)$. The choice of direction at each step depends solely on the relative magnitudes of $ f(k-1, j)$, $ f(k, j-1)$, and $ f(k-1, j-1)$; the absolute value of $ C(B_k, A_j)$ does not affect which predecessor is selected. Consequently, $ C(B_k, A_j)$ should be included in equations (1)–(3), since all three directions represent potential optimal paths. This allows $ C(B_k, A_j)$ to be factored out and added after taking the minimum over the three candidate predecessors, which is consistent with the formulation in traditional dynamic programming.
>
> We recognise that our presentation may cause confusion, so we will add an explicit derivation and a visual dynamic-programming matrix in the appendix to make the alignment path clearer for readers.
>
> **Q3: Regarding Token Order Consistency in Recursion Function and Figure 2b(1).**
>
> Thank you for your insightful observation. Your understanding is correct: in recursion function (1), the two sequences of tokens represent the same text with different tokenizations. Furthermore, due to the dynamic programming definition, the direction can only come from $(k − 1, j − 1)$, $(k, j − 1)$, or $(k − 1, j)$, meaning that token matching can only occur between adjacent positions in order. Therefore, in Figure 2b, there should not be X‑shaped matching (i.e., (1,2)* and (2,1)*), but only I‑shaped (i.e., (1,1) and (2,2)), V‑shaped (i.e., (1,1) and (2,1)), and inverse V‑shaped (i.e., (1,1) and (1,2)) matchings.
>
> *We use (x, y) to represent the token pairing between x: source model token position and y: target model token position.
>
> We apologize again for the confusion caused by the figure and sincerely appreciate your careful reading and for pointing this out. We will revise it accordingly to eliminate any ambiguity.
>
>
> **Q4: Regarding Computational Efficiency of Token Alignment and Optimal Transport.**
>
>
> This is a great question. We fully agree with the reviewer that fully disclosing the computational burden could greatly benefit the community. As detailed in Appendix S2, FuseLLM takes approximately 3 GPU hours (on an NVIDIA A100-80GB) to generate source and target model predictions, and 4 CPU hours (on an AMD EPYC 7763 processor) to conduct token alignment. Compared to FuseLLM, PTA-LLM introduces an additional optimal transport computation, which results in approximately a 13.75% increase in CPU alignment time on the MiniPile dataset. It would be valuable to explore more efficient alternatives, such as the Greenkhorn algorithm, to reduce the complexity of the optimal transport step.
>
> **Q5: Regarding the Task Selection in Stability Analysis**
>
> Thank you for your insightful comment. Following your suggestion, we provide additional comparison results for cases where FuseLLM performs well while PTA‑LLM performs poorly.
>
> | Task                     | LLaMA‑2 | FuseLLM | PTA‑LLM |
> | ------------------------ | ------- | ------- | ------- |
> | Multistep Arithmetic Two | 0.80    | 4.80    | 4.00    |
> | Temporal Sequences       | 12.80   | 16.40   | 15.20   |
>
> As shown in the table above, PTA‑LLM exhibits slight performance degradation in two BBH subtasks, both of which evaluate EM accuracy on various reasoning scenarios. PTA‑LLM applies OT to balance the probability mass in each row, producing a soft assignment matrix that yields more compact and consistent token representations. This smoothing flattens peaks for high‑confidence tokens. In exact‑match BBH tasks, spreading probability to distractor tokens can shift the argmax and reduce accuracy. PTA‑LLM is penalized for minor surface deviations, while FuseLLM’s hard alignment preserves high‑confidence tokens, yielding more frequent matches with the official answers. This explains why PTA‑LLM achieves the smallest accuracy improvement under BBH in Table 2.
>
> We will supplement more comprehensive per‑task results between FuseLLM and PTA-LLM in Appendix S8 to ensure the methods are evaluated under unbiased conditions.
>
> **Q6: Regarding the Figure 3**
>
> We are happy to restate and elaborate on our motivation for presenting Figure 3 and Section 4.4, as they strongly support the core motivation of PTA‑LLM.
>
> The motivation of PTA‑LLM is to achieve a coherent fused metric, as described in our introduction. We aim to keep the logit distribution more compact and consistent with the target distribution, which motivates the design of probabilistic token alignment under the optimal transport framework.
>
> In addition to the effectiveness and stability results shown in Sections 4.2 and 4.3, we incorporate empirical visualizations to support our motivation. As shown in Figure 3, we visually compare the distribution of PTA‑LLM fused tokens with the target tokens and the FuseLLM fused tokens. Specifically, Figures 3(b) and 3(d) show that PTA‑LLM achieves a more consistent marginal feature distribution with the target tokens, whereas FuseLLM exhibits significantly greater distortion in the overall token representation. The more compact and coherent token distribution after applying probabilistic token alignment is consistent with the motivation described above.
>
> We will include this elaboration and add a clarifying note in the revision.
>
> **Q7: Regarding the Equation 4**
>
> Thank you for your great question. We would like to provide further clarification here.
>
> Without constraints on m and n, the transport space between the logits of Model A and Model B would equal the product of their vocabulary sizes (for example, MPT has 50,277 tokens while LLaMA has 32,000). This is computationally **inefficient** because it requires storing over 30k logits per token instead of just the top 10, and it is **unnecessary** because the logit distribution follows a long‑tail pattern in which the top‑k logits capture the majority of the probability mass. Our further analysis of Token Alignment Window Sizes, ranging from 10 to 3 (Tables 4 and S10), shows that a small window size is sufficient for effective model knowledge fusion.
>
> We will include this elaboration and add a clarifying note in the revision.
>
>
> We appreciate your thoughtful comments. We hope our response addresses your concerns. Please let us know if there are any additional questions, and we will be happy to discuss further.

---

> > ### Comment · Reviewer_zYWa · 2025-08-01
> >
> > Thank you for your response. My concerns are addressed. I will keep my positive rating to this paper.

---

> > > ### Author Response · Authors · 2025-08-01
> > >
> > > We’re glad our responses have addressed your concerns and received your approval of our work. We are always ready to respond to any further questions you may have, and we deeply thank you for your thoughtful support in improving our work.

---

### Official Review · Reviewer_sCqG · 2025-06-28

**Clarity:** 4
**Significance:** 2
**Originality:** 3
**Rating:** 5
**Confidence:** 4

**Summary:**

This work introduces a token alignment method for knowledge fusion, designed to merge multiple source LLMs into a single target LLM.

The method makes three main contributions:

1.  It allows for flexible, **many-to-many mappings** between the tokens of the source and target models. This avoids the common issue of requiring a rigid, one-to-one mapping between tokenizers that have vastly different vocabulary sizes, training data, or training algorithms. The output of this stage is a sequence of matched token pairs.

2.  It aligns the **next-token probability distributions** for each token pair, rather than just aligning tokens based on string similarity. This process is framed as an **optimal transport problem** where the goal is to "transport" the probability from the source model's predictions to the target's with minimal cost. The transport cost considers the edit distance between potential predictions, while the transport itself preserves the probability mass between the two distributions. This is solved using the standard Sinkhorn algorithm.

3.  When fusing multiple source LLMs, it uses a recursive merging algorithm that has **linear complexity** with respect to the number of models and prioritises higher-quality models based on a heuristic, avoiding a potential combinatorial explosion.

When used in a knowledge fusion pipeline, these modifications yield small but significant performance gains across a wide range of common downstream tasks.

**Questions:**

- The knowledge fusion pipeline requires running many models to generate training data, which is computationally expensive and would be almost untenable at scale for modern datasets with tens of trillions of tokens. If one uses such a dataset, how should they choose how much to subsample the data in order to use your method? Could they use it for a small targeted subset, such as FineMath?

- Your work focuses exclusively on the knowledge fusion paradigm. Why was there no empirical comparison to other common methods like fusing hidden states via summing/averaging and then appending new blocks afterwards or alternative stitching or model merging methods? A comparison, even on a smaller scale, would be critical for understanding the trade-offs of your method.

- The optimal transport cost function is based on edit distance. Have you experimented with using semantic similarity (e.g., from token embeddings before the unembedding matrix) as the cost metric instead? Previous works, such as [1,2] have shown that it may be feasible to translate between the latent spaces of two models and take distances there instead of using string distance.

Citations:
- [1] Moschella et.al; "Relative representations enable zero-shot latent space communication"
- [2] Alvarez-Melis et.al; "Gromov-Wasserstein Alignment of Word Embedding Spaces"

**Ethical Concerns:**

["NO or VERY MINOR ethics concerns only"]

**Final Justification:**

The author's rebuttal has addressed a majority of my concerns while opening up new avenues of exploration (using hidden representations instead of logits). While I have reservations regarding the practical applicability of the method at scale (specifically concerning serving and storage), it is a compelling scientific exploration and will serve to inform other members of the community.

**Limitations:**

The authors thoroughly discussed the limitations of their work.

**Paper Formatting Concerns:**

No concerns.

**Quality:**

2

**Strengths And Weaknesses:**

### Strengths:
* The work is well-written, with clear explanations and a logical flow.
* The proposed methods are well-justified and effectively address the core challenges of knowledge fusion pipelines. The reformulation of token alignment as an optimal transport problem is a principled approach.
* The authors provide ample empirical evidence to validate the downstream performance improvements, demonstrating the method’s reliability.
* The token pairing and alignment techniques are generalizable. They could be transferred to other machine learning domains that require aligning outputs, such as early-exit models or multi-modal models that rely on discrete tokenizations.

### Weaknesses:
* While consistent, the magnitude of the performance improvements is small in absolute terms.
* The baseline knowledge fusion framework is inherently expensive because it requires obtaining predictions from N models to create the fused training data. This high upfront cost may be a significant drawback for modest performance gains.
* The paper’s scope is narrowly focused on improving a specific paradigm—knowledge fusion—and does not compare its method against other prominent model combination techniques. For instance, there are no empirical comparisons to alternative approaches like model stitching and model merging.
* While the goal is to handle heterogeneous LLMs, a large majority of trained LLMs are based on a GPT-2-like structure with highly similar vocabulary sizes. It would have been beneficial to see controlled experiments with very different architectures. If insufficient diversity is available in the wild, training very small models with different structures would be a sufficient proxy.

---

> ### Author Rebuttal · Authors · 2025-07-31
>
> Dear reviewer sCqG,
>
> We sincerely appreciate your time and effort in reviewing our paper and providing valuable comments. We provide explanations to your questions point-by-point in the following.
>
> **Q1: Regarding the improvements.**
>
> Thank you for your insightful observation. Regarding performance, while our average absolute gain over CLM is 1.1%, we argue that the relative gain of 3.94% is substantial. Furthermore, our significance test results confirm that these improvements are **statistically significant**, with a p-value < 0.005.
>
> Our visualization results further support this claim. The **noticeable distortion** in token representations, as shown in Figures 3 and 6, highlights the **limitations of FuseLLM**. In contrast, the well-preserved representations in PTA-LLM demonstrate its strong potential for broader and more general model fusion scenarios.
>
> **Q2: Regarding Computational Cost of Token Alignment.**
>
> This is an excellent question. We fully agree with the reviewer regarding the concern about the additional burden of token alignment. To further demonstrate the effectiveness of our method, we conducted an additional experiment in Appendix S7.2. to compare two distinct settings.
>
> Setting A: Train on a subset of MiniPile (100K examples by randomly sampling 10% of the original dataset) using the same setting in the paper. Specifically, the total training costs, including obtaining the probability distributions of all teacher models (approximately 3.2 GPU hours) and end-to-end training (approximately 20 GPU hours), are 23.2 GPU hours.
>
> Setting B: Use the additional cost to train on more data. Specifically, we train Llama-2 CLM on a larger MiniPile subset (116K training examples, representing 11.6% of the original dataset) to match the total training costs of Setting A (23.2 GPU hours).
>
> The results, summarized in the table below, are consistent with our findings, demonstrating the benefits of the knowledge fusion paradigm under equivalent resource constraints. We sincerely appreciate your constructive suggestions.
>
> | Same Cost | Setting A (Our method) | Setting B (CLM with more data) |
> | -------------- | ---------------- | ------------------------------ |
> | MMLU           | 47.42            | 46.33                          |
>
>
> **Q3: Regarding Lack of Comparison with Other Model Combination Techniques**
> Thank you for your excellent question. Most traditional fusion methods, such as model merging and fusing hidden states, require identical model architectures and are therefore not directly applicable in our context, in which we fuse heterogeneous LLMs.
>
> However, we further compare PTA‑LLM with an ensemble method for LLMs, LLM‑Blender [ref1], which ranks and combines the output texts from multiple LLMs (and can be applied in heterogeneous LLM fusion) using ranker and fuser models.
>
> | Model                   | BBH  |
> | ----------------------- | ---- |
> | OpenLLaMA               |  33.87    |
> | MPT                     |   33.38   |
> | Llama-2                 |   39.70   |
> | LLM-Blender (Rank&Fuse) |   24.48   |
> | LLM-Blender (Rank)      |  37.17    |
> | PTA-LLM                 |  41.08    |
>
> From the results above, we observe that PTA‑LLM still achieves superior performance. We will include these results in our revision to enhance the quality of our paper.
>
> [ref1] LLM-Blender: Ensembling Large Language Models with Pairwise Ranking and Generative Fusion
>
> **Q4: Regarding Model Fusion Settings.**
>
> Thank you for the excellent suggestion. We fully agree with the reviewer that incorporating more fusion settings would strengthen confidence in the robustness of the method. Therefore, we have indeed included experimental results on model fusion with additional models, including Mixtral and Qwen2.5, in Appendix S7. As shown in these tables, both “PTA-LLM + Qwen2.5” and “PTA-LLM + Mistral” consistently outperform Qwen2.5 and Mistral, respectively, demonstrating the effectiveness of our approach within the knowledge fusion paradigm. We plan to explore fusion with a larger number of models in the future.
>
>
> **Q5: Regarding Scalability and Data Subsampling in Knowledge Fusion.**
>
> This is a great question. We understand the reviewer’s concern that the scalability of real‑world datasets is quite different from the 1M‑sized MiniPile used in our training. Therefore, determining how to effectively subset the data becomes an important problem. We further investigate subsample training in Appendix S7.3. The results show that random sampling from the original dataset demonstrates both robustness and effectiveness for PTA‑LLM. We will incorporate this discussion into our revision to improve the quality of our paper.
>
> **Q6: Regarding Use of Edit Distance vs. Semantic Similarity in Transport Cost.**
>
> Thank you for your insightful suggestion. Our motivation for using minimum edit distance as the evaluation metric comes from the scale of our 1M‑sample training corpus, where even a small increase in complexity could create a significant computational burden for token alignment. A semantic similarity‑based token‑level metric could indeed embed more comprehensive and reasonable information for transportation.
>
> We are also very interested in this direction and plan to investigate it further, exploring more efficient semantic similarity approaches that are well‑suited to our setting.
>
> We appreciate your thoughtful comments. We hope our response addresses your concerns. Please let us know if there are any additional questions, and we will be happy to discuss further.

---

### Official Review · Reviewer_8QeG · 2025-07-02

**Clarity:** 3
**Significance:** 3
**Originality:** 3
**Rating:** 5
**Confidence:** 3

**Summary:**

This paper addresses the problem of token alignment and determining the probabilities of tokens in a situation when one wishes to fuse token probabilities from one or more language models.

If I understand correctly, the main innovation is the fused probability assignment after token alignment. The paper proposes to first find a transport plan between two distributions over vocabularies of different sizes. The fused token probabilities are determined based on this plan.

**Questions:**

The assignment in Equation (5) is a key. However, it appears to have no clear interpretation in transport theory so maybe a better discussion or another visual can help the reader appreciate this key point.

**Ethical Concerns:**

["NO or VERY MINOR ethics concerns only"]

**Final Justification:**

The rebuttal settled several questions and uncertainties that I had.
I think that this is a good paper that should be accepted, as my initial rating says. I did not see room to improve this rating further.

**Limitations:**

Yes

**Paper Formatting Concerns:**

No concerns

**Quality:**

3

**Strengths And Weaknesses:**

Strengths:
- The problem is crisp with a clear motivation.
- Visuals are mostly neat and helpful.
- Experimental results demonstrate the advantage of the method. The methodology seems comprehensive.

Weaknesses:
- It is unclear to me why mixing with CLM loss. Shouldn't model fusion only consider merging logits from the models?
- The challenge in token pairing (Lines 135-146) is a bit unclear. The authors state that the challenge lies in the large search space, but to me, it seems that the main issue is ranking different pairings, as there is no immediate criterion for a good pairing.
- I also identified several writing issues.
    - The concepts of source and target models arise somewhere in Line 40, but it is unclear how these concepts relate to the problem of model fusion discussed previously. Indeed, the discussion so far on model fusion appears to be that we take information from different models symmetrically.
    - It is unclear if the transport is over logits or probabilities. I assume the latter, but this is not clear from the description.
    - In Line 117, the "target's model predictions" is Q_t, but in Line 125 you use P_t for the target model's predictions. There appears to be confusion.

---

> ### Author Rebuttal · Authors · 2025-07-31
>
> Dear reviewer 8QeG,
>
> We sincerely appreciate your time and effort in reviewing our paper and providing valuable comments. We provide explanations to your questions point-by-point in the following.
>
> **Q1: Regarding the Use of CLM Loss in Model Fusion.**
>
> Thanks for the great question, and we'd like to provide further explanation.
>
> When we fuse models through “knowledge fusion,” the process is similar to distillation, transferring the fused logits into the student model. However, the student still needs to learn to predict the actual next token on real data, similar to standard LLM distillation. Specifically, we ensure that the fused model continues to align with real data while absorbing knowledge from multiple teachers. Without the CLM term, there is a risk that the model will merely echo the fused logits and lose fidelity to the true next‑token distribution. Incorporating CLM anchors the fusion to real‑world performance.
>
> **Q2: Regarding the True Challenge in Token Pairing.**
>
> Thank you for your insightful observation. We would like to further clarify and elaborate on the token pairing process. Specifically, the **first** step is to establish a ranking criterion for determining the optimal pairing. We use minimum edit distance as an efficient evaluation metric, given the 1M‑sample training corpus. **Then**, once a reasonable criteria is defined, the main challenge shifts from how to pair to how to pair it efficiently. Dynamic programming reduces the exponential path enumeration to $ O(LN)$, ensuring that pairing remains tractable even for sequences with thousands of tokens.
>
> We will refine the relevant writing in the revised version to ensure better coherence and enhance overall clarity.
>
> **Q3: Regarding the writing.**
>
> **Ambiguity in Source and Target Model Definitions:**
>
> Sorry for the confusion. The definitions of source and target Model are intended to clarify our final base model (the target model) and the direction of knowledge flow (from source to target). As illustrated in Figure 1(c), knowledge from the “cat” and “rabbit” source models is fused into the “llama” target model. We will revise the text to enhance the clarity of our paper.
>
> **Transport objective:**
>
> Thank you for your insightful comment. As detailed in Appendix S5, we perform optimal transport on logits after applying the softmax function (i.e., probability) to reduce the impact of extreme values (e.g., extreme large, small, or negative values) that could otherwise distort the transport cost. We retain the term ‘logit’ for uniform notation in our paper. Importantly, conducting transport in logit space differs fundamentally from transporting mass in probability space due to the distinct normalization terms associated with the source and target spaces. We plan to investigate i further.
>
> We will move this discussion from appendix to the main paper for better clarification.
>
> **Notational Inconsistency in Target Model Predictions:**
>
> Thank you for pointing this out. This is a typo, and it should be $Q_t$ in line 125. The reason we use two notations is to distinguish between the alignment matrix $P_f$ and the final training matrix $Q_t$. The former is obtained from token alignment (without gradient updates), while the latter represents the logits output (with gradient updates) used to align $P_f$ and $O_t$.
>
> **Q4: Regarding Interpretation and Clarity of Equation (5) Assignment**
>
> Thank you for your excellent question. We would like to elaborate on the motivation behind Equation 5 and its empirical visualization.
>
> **Motivation:** To achieve a coherent fused metric as described in our introduction, we aim to keep the logit distribution more compact and consistent with the target distribution. Due to the constraints in optimal transport, we discretely select the maximum value in each sub‑transport, which helps maintain both compactness and consistency. We are also interested in whether broader transport (e.g., choosing top‑k instead of top‑1) could further improve fusion, and we plan to investigate this in future work.
>
> **Empirical Visualization:** As shown in Fig. 3, we can visually compare the distribution of PTA‑LLM fused tokens with the target tokens and the FuseLLM fused tokens. Specifically, Fig. 3(b) and Fig. 3(d) show that PTA‑LLM achieves a more consistent marginal feature distribution with the target tokens, whereas FuseLLM exhibits significantly greater distortion in the overall token representation. The more compact and coherent token distribution after applying probabilistic token alignment is consistent with the motivation described above.
>
> We will include this elaboration and add a clarifying note in the revision.
>
>
> We appreciate your thoughtful comments. We hope our response addresses your concerns. Please let us know if there are any additional questions, and we will be happy to discuss further.

---

> > ### Comment · Reviewer_8QeG · 2025-08-05
> >
> > Thank you for your response.
> >
> > I am satisfied with the clarification and the answer to my question.

---

> > > ### Author Response · Authors · 2025-08-05
> > >
> > > We’re glad our responses have addressed your concerns. We are always ready to respond to any further questions you may have, and we deeply thank you for your thoughtful support in improving our work.

---

### Official Review · Reviewer_seY2 · 2025-07-03

**Clarity:** 3
**Significance:** 2
**Originality:** 3
**Rating:** 4
**Confidence:** 3

**Summary:**

The paper aims to tackle the problem of vocabulary alignment in knowledge fusion across models. The method proposed seeks to obtain a probabilistic token alignment using optimal transport theory, in contrast with past work where the mapping was hard. Additionally, the proposed method for alignment leverages the top-k predictions. Experiments in merging three models show that the resulting fusion method obtains better results than any base model, continuously pre-training the base model and a fusion method that uses hard mappings.

**Questions:**

n/a

**Ethical Concerns:**

["NO or VERY MINOR ethics concerns only"]

**Final Justification:**

The rebuttal provided information I already had when making my original review, so I will maintain my overall opinion, which is borderline accept.

**Quality:**

3

**Strengths And Weaknesses:**

Strenghts:
- original idea to use optimal transport for token alignment. Based on previous work, this is a natural evolution in this area of research
- the problem is well motivation and theoretically sound
- ablation experiments
- study of interpretability

Weaknesses:
- experiments are not varied enough i.e. only a single setup is presented with 3 models to fuse from. I would expect at least 2 merging setups, ideally with models to merge from different families, and with models of different sizes. This would improve confidence in the robustness of the method
- results show improvements over the CLM setup, but the improvements when compared to FuseLLM, which is the most directly related work, are quite small (+0.49 on average, but gains on 3/6 datasets are 0.32 or smaller.

---

> ### Author Rebuttal · Authors · 2025-07-31
>
> Dear reviewer seY2,
>
> We sincerely appreciate your time and effort in reviewing our paper and providing valuable comments. We provide explanations to your questions point-by-point in the following.
>
> **Q1: Regarding Model Fusion Settings.**
>
> Thank you for the excellent suggestion. We fully agree with the reviewer that incorporating more fusion settings would strengthen confidence in the robustness of the method. Therefore, we have indeed included experimental results on model fusion with additional models, including Mixtral and Qwen2.5, in Appendix S7. As shown in these tables, both “PTA-LLM + Qwen2.5” and “PTA-LLM + Mistral” consistently outperform Qwen2.5 and Mistral, respectively, demonstrating the effectiveness of our approach within the knowledge fusion paradigm. We plan to explore fusion with a larger number of models in the future.
>
> **Q2: Regarding the Improvements.**
>
> Thank you for your insightful observation. Regarding performance, while our average absolute gain over CLM is 1.1%, we argue that the relative gain of 3.94% is substantial. Furthermore, our significance test results confirm that these improvements are **statistically significant**, with a p-value < 0.005.
>
> Our visualization results further support this claim. The **noticeable distortion** in token representations, as shown in Figures 3 and 6, highlights the **limitations of FuseLLM**. In contrast, the well-preserved representations in PTA-LLM demonstrate its strong potential for broader and more general model fusion scenarios.
>
> We appreciate your thoughtful comments. We hope our response addresses your concerns. Please let us know if there are any additional questions, and we will be happy to discuss further.

---

### Author Response · Authors · 2025-08-09
**Summary of Rebuttal and Discussion**

Dear Area Chair and Reviewers,

We would like to express our sincere gratitude for your efforts in facilitating the discussion regarding our paper. We are glad that **we received all reviewers’ approval during the rebuttal**, that they maintained their positive assessments, and that Reviewer sCqG has even raised the rating in recognition of the quality of our paper and rebuttal. Although we understand that Reviewer seY2 has not engaged in subsequent discussion due to a busy schedule, we believe that Reviewer seY2’s acknowledgement indicates that our response has effectively addressed the concerns through clear explanations.

---

As the discussion is coming to an end, we would like to provide a brief summary of the key points that have been discussed:

- As suggested by Reviewer sCqG, we have supplemented our work with **an additional comparison to the paradigm of model ensembles**, given that other paradigms can not be applied under our heterogeneous fusion settings, underscoring the superiority of PTA-LLM.
- In response to Reviewers seY2, 8QeG, and sCqG regarding more diverse fusion settings with various LLM families, and to Reviewer zYWa regarding computational efficiency, we have **clarified and included the findings from experiments already conducted in our appendix** to address their concerns.
- Regarding the writing improvements pointed out by Reviewers 8QeG, sCqG, and zYWa, we have provided detailed explanations of our motivations, implementation details, mathematical equations, the overall training objective, and the visualizations. We are **committed to refining our work** based on the feedback received.
- We especially appreciate the detailed comments and suggestions from Reviewer zYWa on task selection improvements and from Reviewer sCqG on using semantic similarity, as these are crucial for improving the quality of our work.

---

We would also like to emphasize the contributions of our work, which have been acknowledged by the reviewers and are important to the community:

- **Unveiling the essence of token alignment through intuitive visualizations**, we support and cross-validate our approach under the framework of optimal transport theory, while clearly elucidating the token alignment process and eliminating the token distortion caused by previous hard-mapping approaches.
- **Investigating the potential of heterogeneous fusion and its linear scalability to a larger number of models (i.e., Qwen and Mistral)**, we highlight its high efficiency and offer valuable insights for constructing effective and efficient future LLMs.
- **Conducting comprehensive experiments across diverse benchmarks covering core LLM capabilities (i.e., detailed case studies and analyses),** we demonstrate the generality of our approach through coherent representation fusion.

Finally, we deeply value the constructive comments provided by the reviewers. In response, we have carefully refined our work based on the feedback received. Considering the contributions made, we hope that our work can provide new insights to the model fusion communities, and contribute to their further development.

Sincerely,

Authors

---

### Decision · Program_Chairs · 2025-09-17

**Decision:**

Accept (poster)

**Comment:**

This paper addresses the problem of combining multiple large language models into a stronger model in a setting in which the token vocabulary differs across models. All the reviewers are positive or mildly positive about the paper, and they were satisfied with the rebuttal by the authors.